# Immediate action is the best strategy when facing uncertain climate change

Maria Abou Chakra [1,2], Silke Bumann[1], Hanna Schenk[1], Andreas Oschlies [3] & Arne Traulsen [1]

Mitigating the detrimental effects of climate change is a collective problem that requires global cooperation. However, achieving cooperation is difficult since benefits are obtained in the future. The so-called collective-risk game, devised to capture dangerous climate change, showed that catastrophic economic losses promote cooperation when individuals know the timing of a single climatic event. In reality, the impact and timing of climate change is not certain; moreover, recurrent events are possible. Thus, we devise a game where the risk of a collective loss can recur across multiple rounds. We find that wait and see behavior is successful only if players know when they need to contribute to avoid danger and if contributions can eliminate the risks. In all other cases, act quickly is more successful, especially under uncertainty and the possibility of repeated losses. Furthermore, we incorporate influential factors such as wealth inequality and heterogeneity in risks. Even under inequality individuals should contribute early, as long as contributions have the potential to decrease risk. Most importantly, we find that catastrophic scenarios are not necessary to induce such immediate collective action.

[1] Department of Evolutionary Theory, Max-Planck-Institute for Evolutionary Biology, August-Thienemann-Straße 2, 24306 Plön, Germany. [2] Donnelly Centre for Cellular and Biomolecular Research, University of Toronto, 160 College Street, Toronto, QC M5S 3E1, Canada. [3] Biogeochemical Modeling, GEOMAR Helmholtz Centre for Ocean Research Kiel, Düsternbrooker Weg 20, 24105 Kiel, Germany. Correspondence and requests for materials should be addressed to A.T. (email: traulsen@evolbio.mpg.de)

Collectively working to manage climate change is one of the most difficult social challenges we face today. It is imperative to find a solution, because failing can mean disaster for our own future and generations to come. However, in spite of the obvious consequences, achieving cooperation at the global level remains a challenge because individuals are tempted to reduce their efforts while taking advantage of everyone else's contributions[1,2]. Such selfish behavior lies at the core of the tragedy of the commons—a classic problem showing how the pursuit of individual gain leads to the overexploitation of a resource[3,4]. Social dilemmas are often modeled as public goods games where individuals decide to contribute to a collective good. Each individual receives a benefit that depends linearly on the total contributions[5–10]. But in light of dangerous climate change, the typical linear public good game is insufficient and, thus, the collective-risk game with multiple rounds, threshold effects and partial returns was devised[11]. Contrary to the usual case of a collective benefit, this game is concerned with the prevention of a collective loss. The collective-risk game is usually played over several rounds. In each round, individuals must decide how much they contribute to a public good. If the total contributions exceed a specified target amount, the risk of a collective loss vanishes. In contrast, failure to reach the target implies that individuals lose all their belongings with some probability. This setup is motivated by dangerous climate change, "the game we cannot afford to lose"[12].

The collective-risk game was dubbed the climate game after stimulating several fascinating experimental and theoretical studies[13–19]. While these studies have enhanced our understanding about how collective losses influence cooperative behavior, they all build on certain limiting assumptions[11,14,19–21]: loss is devastating because individuals are deprived of all their possessions, risk is typically characterized by threshold effects, and a single potential loss occurs at the end of the game. However, real global scale environmental crises that may be triggered by climatic events—ranging from devastating heat waves, storms, floods or droughts over shifts in monsoonal precipitation patterns to a collapse of parts of the Antarctic or Greenland ice sheets—are currently unpredictable with respect to their impact and timing. Also, in the context of climate change, there is an ongoing discussion about predicting when and how drastic the effects will be[22,23], as well as the advantage of intermediate climate targets[17,24]. These real world examples represent situations where disasters are unpredictable and possibly recurring, thus making it difficult to assess how players should behave in these situations.

To shed further light on cooperative behavior given such uncertainty, we develop a collective-risk game where multiple losses can occur—similar to the experiment by Milinski et al.[17] where losses may happen in every round when an intermediate target is missed. Having a potential loss in every round also introduces situations where losses are not devastating, i.e., individuals do not lose everything. In contrast to the typical single catastrophic event that leads to full losses and the sudden termination of the game[25], we further explore the effects when a loss occurs at a fixed time, such as the start or end of a game, or at a random time, where individuals do not know if and when a disaster will occur. In addition, we consider factors that play a key role in decision making: wealth inequality, heterogeneity in risk probability and heterogeneity in the distribution of individuals within a group. In such collective-risk dilemmas, heterogeneity, risk, and coordination play a large role in decision making[15,17,26,27]. Most often, heterogeneity renders negative outcomes. Since nations and individuals vary on several factors, it is difficult to assess which of these factors has the largest influence on climate change decisions. For instance it is unclear how individuals should interact under wealth inequality: previous work has shown that the amount of contributions also depends on the shape of the risk function[9,26,28] or communication[15,26]. Another problem under wealth inequality is determining which distribution of efforts can be considered as fair[29].

Using this general setup, we study the amount as well as the timing of contributions under various risk scenarios. It turns out that the risk scenario at stake has crucial effects on the evolutionary optimal behavior in the game and in particular also on the timing. While wait and see is a favorable strategy when the potential loss happens at a well known time, an act quickly behavior is more successful under uncertainty and repeated threats.

## Results

**Model.** We devise a game where the risk of a collective loss can be unpredictable or can recur. The game is played by a group of $m$ individuals over $\Omega$ rounds. Initially, each individual $i$ in the group has a wealth, $W_{i,0}$. In each round, $r$, individual $i$ contributes $c_{i,r}$ from $W_{i,r-1}$ to a collective pot, $C_r$, that represents the public good. To model collective loss, we assume that in each round a fraction $\alpha_i$ is removed from the remaining wealth of individual $i$ with probability $p_i[C_r]$, which depends on $C_r$, the total contributions so far. Initially, we explore the homogenous case where $W_{i,0}$, $p_i[C_r]$ and $\alpha_i$ are identical for all players. Heterogeneity arises when we change these parameters among the players, for example under wealth inequality, $W$ differs among players. For simplicity we only consider two types, rich and poor individuals, and explore the cases where $W_R \geq W_P$. We also explore heterogeneity in the loss fraction $\alpha_i$, where individuals can lose different portion of their wealth in a loss event. We also combine factors such as wealth inequality and risk heterogeneity: where risk probability $p_i$ could differ between populations of rich and poor individuals, such that the risk curves and thus the contributions depend on the wealth status ($p_R[C_r] \neq p_P[C_r]$).

The probability to lose is monotonically decreasing in the total contributions of the individuals in the group over all rounds so far, $C_r = \sum_{j=1}^{r} \sum_{i=1}^{m} c_{i,j}$, and is captured by the risk curve $p_i[C_r]$. In the event of a loss in round $r$, an individual keeps the amount of wealth $W_{i,r} = (1 - \alpha_i)(W_{i,r-1} - c_{i,r})$ after that round. With probability $1 - p_i[C_r]$ an individual evades the risk and keeps $W_{i,r} = W_{i,r-1} - c_{i,r}$. The remaining wealth can be used to contribute to the collective pot in the next round. By the end of the game, the expected payoff is calculated. For example, in a two-round game: if a loss happens in round one, the remaining wealth of a player $i$ after the first round is $W_{i,1} = (W_{i,0} - c_{i,1})(1 - \alpha_i)$. In round two, the player can then still contribute $c_{i,2} \leq W_{i,1}$, leading to a payoff of $(W_{i,0} - c_{i,1})(1 - \alpha_i) - c_{i,2}$ if no second loss happens (in the following, we implicitly assume that $0 \leq c_{i,r} < W_{i,r-1}$). There are four possible outcomes since the events with loss and without loss can happen independently in each round. The expected payoff is given by

$$\pi_i = (1 - \alpha_i p_i[C_2])\Big((1 - \alpha_i p_i[C_1])\big(W_0 - c_{i,1}\big) - c_{i,2}\Big),$$

where $p_i$ is a function of the total contributions made over all rounds within the group so far, $C_r$, and thus also of $c_{i,1}$ and $c_{i,2}$.

In our model, contributions help avoid a collective loss in each future round of the game. Contributions made early on in the game are not in vain, as they reduce the risk of events leading to a loss in future rounds—but they cannot recover what has already been lost in earlier rounds. Thus, it would be socially optimal to contribute as early as possible and to distribute contributions evenly among players[30]. However, herein we consider players who are only interested in their own individual advantage instead. We apply evolutionary game theory[14,19,20,31–33] to understand and identify the set of stable contributions under various risk

scenarios. This implies that we focus on stationary solutions of the behavior dynamics in a large population (typically 100 individuals) from individuals interacting in groups within a game (to disentangle group effects from risk effects we first focus on the pair-wise case, $m = 2$. Qualitatively similar effects are seen in larger groups, $m > 2$, (Supplementary Fig. 1)). Evolutionary stability implies that a player with an altered contribution scheme would have a lower payoff and thus be less successful[34]. In the case of the rich and a poor scenario, the evolutionary processes are independent—we assume two distinct populations. This setup ensures that poor players will preferentially adopt behaviors that have been beneficial for other poor players, but not try to imitate the behaviors of rich players (and vice versa).

The emerging contribution scheme depends on the fraction of wealth that is lost, $\alpha$, the shape of the risk curve, $p[C_r]$, and the number of rounds $\Omega$. We consider the influence of the fraction lost, $\alpha$, for different risk curves (Fig. 1a): linear or piece-wise linear curves, as well as non-linear curves with threshold effects. While linear curves represent particularly simple cases, threshold effects might be especially relevant when it comes to the earth's climate[22].

**Model analysis**. It is clear that risk can influence cooperative behavior in various ways. However, in reality the precise shape of the risk curve is unclear. For example, climate scientists generally do not agree on the exact position of thresholds and how steep the change in risk is[22–37]. To take this ambiguity into account, we vary the shapes of our risk curves. This also allows us to better understand how certain risk curve characteristics affect contributions. For example, how are contributions influenced by sudden changes in risk or by risk that does not decrease much with collective effort?

High values of $\alpha$ imply that contributions are beneficial because large potential losses can be avoided. Consequently, contributions tend to increase with $\alpha$ (Fig. 1b, c). At high $\alpha$ individuals contribute approximately half of their endowment in two-round games with linearly declining risks. They tend to give slightly more if threshold effects are present. Moreover, individuals succeed in eliminating risk almost entirely when this can be achieved with relatively little effort. If $\alpha$ is small, losses are minor—individual contributions, therefore, remain relatively low. Intermediate values of $\alpha$ lead to differentiated amounts of contributions, because of the complex interactions between the fraction of wealth lost and the risk curve. This effect becomes especially apparent in the four-round game (Fig. 1d). Moreover, the longer the game lasts, the more individuals contribute at intermediate values of $\alpha$—it can even make sense to contribute less if $\alpha$ increases, as some risk curves demand such high contributions that the remaining wealth becomes negligible and it is beneficial to maintain higher risks. However, as the number of rounds increases, intermediate losses may accumulate over the course of the game, adding up to a large overall loss. The effects of timing have only recently become apparent, experimental evidence and theoretical framework all pointing to the trend that increasing round number increases cooperation[21,35,36].

In the case of linear risk curves and multiple rounds, intermediate-sized losses elicit sizable contributions (for linear risk curves with very steep slopes, contributions are reduced accordingly (Fig. 1). Similar effects on contributions occur using power function risk curves. Furthermore, linear and power function risk curves illustrate that contributions can be largest at intermediate values of $\alpha$, if contributions have only small effects on the probability to lose, or, if it requires a lot of effort to reduce risk. Intuitively, the prospect of having nothing left depresses cooperative effort when risk is high. By comparison,

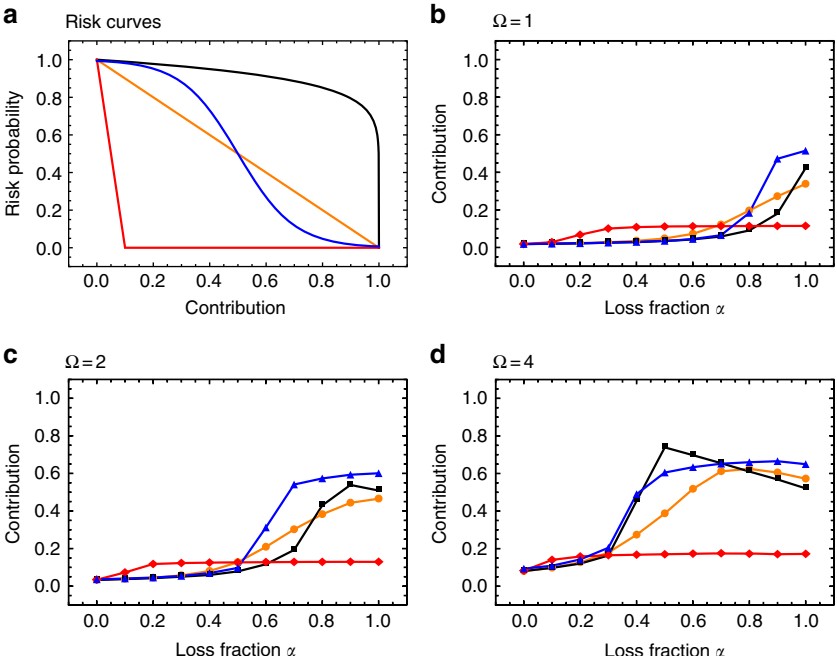

**Fig. 1** Risk curves and the effect of the fraction lost on contributions. **a** Four different risk curves are explored: linear (orange), piece-wise linear (red), a power function (black), and curves exhibiting threshold effects (blue). In the remaining panels, the average contributions for different values of the fraction lost, $\alpha$, are shown. **b** In the collective-risk game with one round ($\Omega = 1$) exclusively large values of $\alpha$ elicit contributions. **c** In the game played over two rounds contributions tend to rise in $\alpha$. **d** Multiple potential losses induce a clear rise in contributions (games are between two individuals, $m = 2$, evolutionary dynamics with a population size $N = 100$ and averages over $10^5$ generations, 1000 games per generation, mutation probability $\mu = 0.03$, and the standard deviation for mutations in the individual thresholds determining contributions $\tau_r$ is set to $\sigma = 0.15$ (Methods section). The functional forms of the curves are given in the Methods section)

contributions for risk curves with threshold are increasing in $\alpha$ (Fig. 1). This curve shows that in the one-round game contributions are low when risk curves start at low risk for low contributions (small $\lambda_3$), but in the four-round game reach those of the curves with high initial risk, provided $\alpha$ is large (Fig. 1d). Interestingly, we find that for the game with multiple losses, high risk probability at low contributions and full losses are not necessary to observe cooperative behavior.

**Unpredictable timing of losses**. So far, we have studied the case of a potential loss in every round of the game, including the possibility of recurrent events. However, the actual timing of catastrophic events associated with climate change is still controversial[22]. Previous empirical and theoretical research has focused on losses in the last round of the game[11,20,21]. These studies have found that low and intermediate loss probabilities lead to a withdrawal of effort by individuals. To put these findings into perspective, we explore the effects of different timings of losses on contributions. In addition to a potential loss event in every round or only in the last round, we now consider scenarios with a potential loss only in the first round or in a single random unknown round within the game. The random round captures, most closely, our current state since the true timing of when climate change will lead to catastrophic losses is unknown[22]. It seems natural to assume that individuals will behave as if climate events hit at random times.

Qualitatively, the game with multiple losses elicits a similar temporal pattern of contributions as the game where the loss occurs in the first round (Fig. 2a, b). The game where the loss event is in the last round elicits a similar amount of contributions as the game where the potential loss event is in a random round—unless there is a threshold effect where players tend to increase their contributions over time when the loss event happens in the last round. However, the timing for these contributions differs (Fig. 2c, d). In the game where a climatic event is fixed to the last round, we recover wait and see behavior[11,20,21] or the Schelling

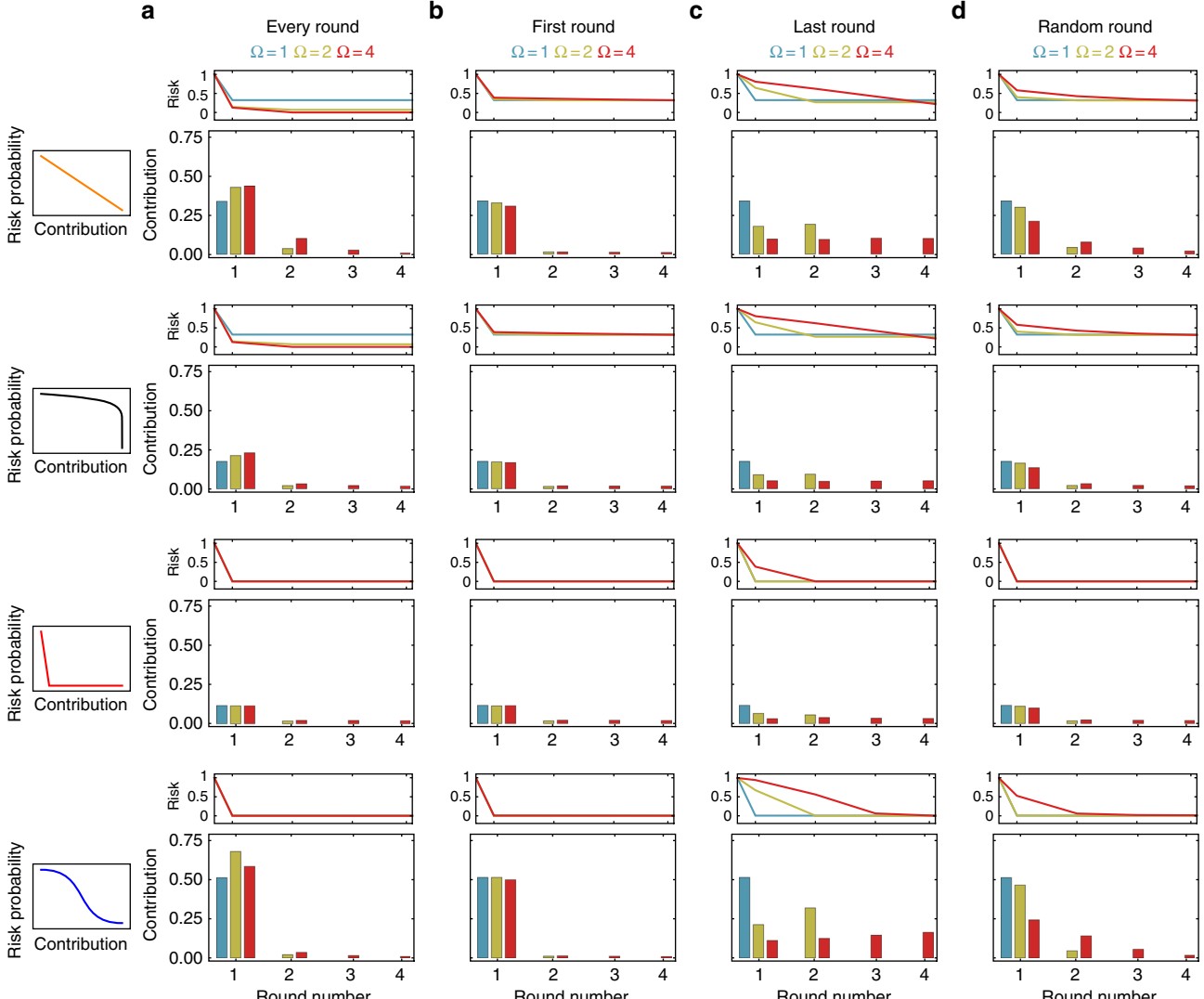

**Fig. 2** Contributions for different timings of potential losses in a game with up to four rounds. In a game with $\Omega$ rounds, for four different risk curves, we show the average contributions of two players (large panels) and the resulting risk probability (small panels) with a potential loss in **a** every round, **b** the first round, **c** the last round, or **d** a random round. It is assumed that individuals lose everything, i.e., $\alpha = 1$ for all individuals. **a** Multiple losses in the four-round game lead to increased contributions, in particular in the first round. **b** If the risk is only present in the first round, no further contributions emerge, as expected. **c** if the risk is only present in the last round[11,20,21], we recover the wait and see behavior if there is a strong threshold effects, see also Methods. **d** The scenario where a single event can hit in a random round is qualitatively similar to the last round scenario (games are pair-wise, population size $N = 100$, 1000 games per generation, mutation rate $\mu = 0.03$, and the standard deviation for mutations in the individual decision thresholds $\tau_r$ is set to $\sigma = 0.15$)

behavior of equal contributions across all rounds[30]. When the occurrence of a loss is predetermined and exhibits strong threshold effects, individuals can time their contributions and only contribute in the round just before the expected loss. Moving beyond this particular case, we find that act quickly is most robust instead. Furthermore, our results show that one should not only act early, but also contribute sufficiently to ensure protection in the future.

**Wealth heterogeneity and risk**. Up to this point we have assumed that the risk perceived by the players is identical. This is far from the current situation where nations are impacted differently from one another, for instance a climate disaster can affect poor nations at a greater magnitude than rich nations. We incorporate influential factors such as wealth inequality and heterogeneity in risks into our evolutionary model, as shown in the Methods section where a complete description of the methods is given and the full range of heterogeneity effects is explored. To put these findings into perspective, we explore the effects of heterogenous loss fractions across different timings of loss events. Intuitively, we expect that low values for the loss fraction $\alpha$ would reduce contributions while higher values should promote contributions irrespective of the heterogeneity between the players. However, the drastic effects of the timing of the events on contributions between rich and poor players was not as intuitive (Fig. 3). Random unpredictable-risk events increased overall contributions from the poor players relative to the game with a single predictable events (Fig. 3a–d). Surprisingly, the games with multiple loss events resulted in higher contributions by the rich player even for low values of the loss fraction $\alpha \leq 0.2$. For high $\alpha$ and for multiple loss events the rich players contributed more than 60% of their initial wealth. Heterogeneity between players only re-emphasized act quickly behavior, which is most robust in the natural cases where multiple losses are possible or the event is unpredictable (Fig. 4a, b). Just as in the homogenous case in the game where a climatic event is fixed to the last round, we recover the Schelling behavior of equal contributions across all rounds, (Fig. 4c)[30]. For all other cases with uncertainty and heterogeneity (Fig. 4a, b, d), we find that act quickly is most robust.

## Discussion

Many unsettled questions about climate change remain. In particular, it is unclear how fast the probability of severe climate change events declines for increasing event amplitudes. An intense event does not have to have a direct effect on losses and may only affect individuals in the distant future. Further uncertainties in the risk probability can manifest because of uncertainties regarding the magnitude of destruction incurred by climate events or uncertainties about the appropriate discounting[38,39]. Taken together, these unsettled grounds give rise to a long chain of structural uncertainties[40]. To reflect some of these uncertainties, we have expanded on the typical assumptions of the literature on collective risk to capture some aspects of our current state. The collective-risk game was developed to study social dilemmas involving risk of a collective loss such as the mitigation of climate change effects[11,12]. A main conclusion drawn from this literature is that a high loss probability promotes contributions[11,20,21]. The collective-risk game became known as the climate game, but it assumed that there is a single loss in the last round of the game and that individuals lose all their belongings. This may be questionable in reality[13–31]. Thus, we have developed a collective-risk game where individuals are faced with the risk of a loss in each round, in a single fixed round or a random round. Another distinct characteristic of our game is that individuals may retain some of their belongings after an event.

Our general setup explores cases where the risk of a collective loss is unpredictable or can recur, and individuals can experience a range of mild and catastrophic scenarios, see Figs. 1 and 5. In contrast to wait and see, which is most successful for a game with a single loss in a fixed round (i.e., the timing of risk is known) and where the risk curve exhibits strong threshold effects, here we obtain an act quickly behavior where individuals reduce the risk by contributing a substantial amount in the first round. This is a striking difference between our current work and previous research on climate cooperation. Moreover, our results show that catastrophic scenarios are not necessary, as we observe contributions as long as individuals can make a difference and change the risk they face.

The framework presented here may bring us a step closer towards understanding the best actions of individuals in real and

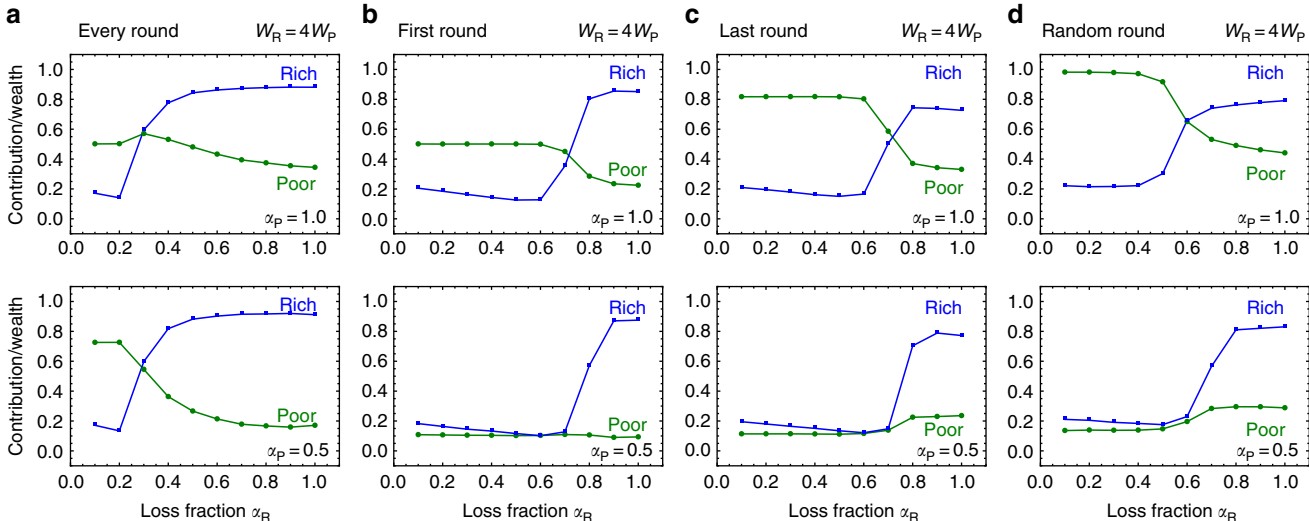

**Fig. 3** Variation of lost endowment for rich and poor player. We plot the average contribution from rich and poor players across various $\alpha_R$ for the rich player in a four-round game where risk is in **a** every round, **b** the first round, **c** the last or **d** a random round. Simulations show when poor players lose everything, $\alpha_P = 1$ or half of their wealth $\alpha_P = 0.5$ (games are pair-wise, $p[C_r] = \left(1 + \exp\left[\lambda\left(\frac{C_r}{W_0} - \frac{1}{2}\right)\right]\right)^{-1}$ with $\lambda_3 = 10$, $W_0 = W_R + W_P$, population size $N = 100$, 1000 games per generation, mutation rate $\mu = 0.03$, and the standard deviation $\sigma$ for mutations of the individual decision thresholds $\tau_r$ is set to 0.15)

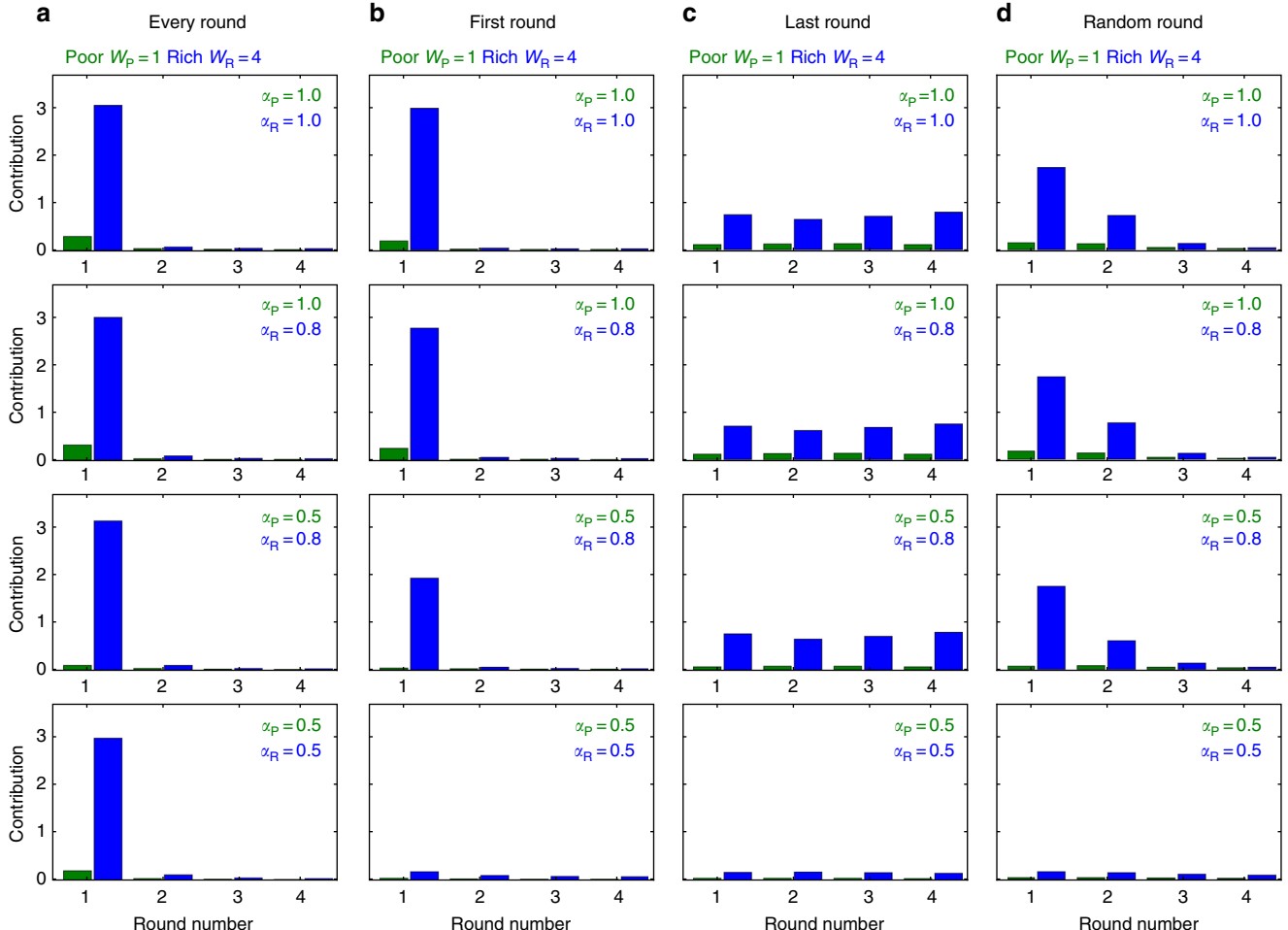

**Fig. 4** Contributions for different timings of potential losses in a four-round game. The graphs depict average contributions in each round of a four-round game with a potential loss in **a** every round, **b** the first round, **c** the last, or **d** a random round (games are pair-wise, $p = \left(1 + \exp\left[\lambda_3 \left(\frac{C_r}{W_0} - \frac{1}{2}\right)\right]\right)^{-1}$, $\lambda_3 = 10$, initial wealth of both players $W_0 = W_{R,0} + W_{P,0} = 4 + 1 = 5$, population size $N = 100$, 1000 games per generation, mutation rate $\mu = 0.03$, and the standard deviation for mutations in the individual decision thresholds $\tau_r$ is set to $\sigma = 0.15$)

often complex situations such as the environmental and global challenges heading our way. In the case with multiple potential losses, failure to act collectively can have severe consequences. By the same token, failure to protect the climate can result in devastating losses. As we have seen in our model, multiple potential losses induce a very different kind of cooperative behavior, Fig. 2. However, multiple losses are not the only factors that affect decisions. One additional factor is group size, see Supplementary Fig. 1, large groups typically reduce cooperation— but we show that timing can quench its effects[21]. Another additional factor is perceived risk, which strongly influences decision making. Our simulations show that wealth inequality can be the factor that may quench risk effects rather than hinder it, as shown in Figs. 3 and 4. Small differences in the risk perceived between parties can lead to negotiation failures for reducing global green house gas emissions; since some parties do not fear climate impact while others have had several drastic climatic events and thus consider risks in their decisions[41]. However, our results indicate that under risk heterogeneity, players with heterogeneous wealth contribute more than player of equal wealth, as observed in Supplementary Figures 3 and 4. Unfortunately, including any form of heterogeneity, such as wealth inequality or risk asymmetries[15,17,26,27,42], can complicate an already difficult situation even further. Differences in wealth can alter decisions because the diverse incentives cause overall uncertainty[38]; wealth inequality is usually associated with negative outcomes[9], however, this also depends on whether subjects felt that this inequality is merited or not. For instance, privileged subjects that feel they were inappropriately endowed equalized their profits with the less fortunate subjects[43]. In contrast, subjects who justified their endowments made no efforts to equalize[44]. Here, we show that even in situations complicated with heterogeneity and uncertainty in the timing of losses, the trends remains the same: it is beneficial for individuals to increase their efforts from the start, ensuring that contributions are sufficient not only for now, but also in the future. Our model thus predicts that immediate climate action can be advantageous at the individual level for a wide range of risk and heterogeneous scenarios.

## Methods

**Simulations**. We use evolutionary game theory to identify evolutionary robust contributions between interacting players[33]. This means players do not know the structure of the game and cannot apply advanced reasoning about their possible actions. Instead, each individual has a fixed mode of behavior, i.e. it follows a certain contribution behavior that only depends on the previous total contributions. Individuals play many games and their success in these games determines how likely it is that their strategy will be adopted by the future generation. The outcome of each interaction depends on the strategy used by each player, which is hard-coded for any individual[20,21]. Each strategy is defined by a threshold, $\tau_r$ (that depends on the collective contributions accumulated over all rounds so far, $C_r$) and the contributions above and below the threshold for each round ($\tau_r$; $a_r$, $b_r$). Thus, an individual will contribute an amount $a_r$ if $C_r \leq \tau_r$ and $b_r$ if $C_r > \tau_r$. As an

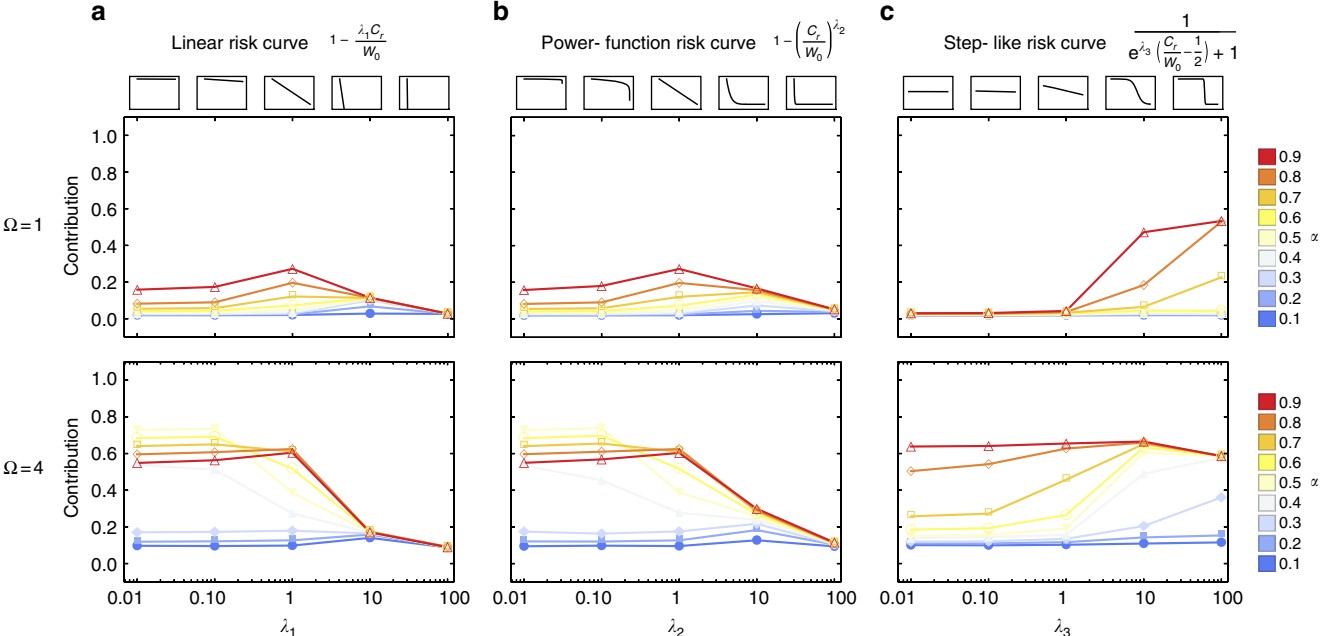

**Fig. 5** Exploring the shape of risk curves. Average contributions for various risk curves and fractions of lost wealth $\alpha$ (see color code) are shown. In case the risk hits, each player, $i$, will lose $\alpha_i W_{i,r}$ for each loss event. The shape of the curve is regulated by parameter $\lambda$ on the x-axis. **a** In the case of linear risk curves and $\Omega = 1$, contributions are largest at intermediate values of $\lambda$ (contributions are small if the curve has a steep slope). **b** Similar effects on contributions apply to the power function risk curve. **c** The step-like risk curve with lower initial risk for smaller values of $\lambda_3$ shows that contributions are rising in $\alpha$ (games are between two individuals, initial wealth of both players $W_0 = 2$, population size $N = 100$, averages over $10^5$ generations, 1000 games per generation, mutation rate $\mu = 0.03$, and the standard deviation for mutations in the thresholds $\tau_r$ is set to $\sigma = 0.15$, see ref. [20] for a similar simulation

example, consider a game with two players and two rounds and no risk, $m = 2$ and $\Omega = 2$. The strategy of player one is {(0.0;0.1,0.0), (0.2;0.1,0.5)}, where the first set of three numbers defines the strategy in the first round and the second set of three numbers defines the strategy in the second round. Player two has strategy {(0.1;0.5,0.1)$_{round1}$, (0.7;0.2,0.5)$_{round2}$}. Since the pot is empty in round 1 ($C_1 = 0$), we have $C_1 \leq \tau_1$ for player one, who thus invests 0.1. For player two, we also have $C_1 \leq \tau_1$, which leads to an investment of 0.5. Thus, in round 2 we have a pot of $C_2 = 0.6$. Consequently, for player one $C_2 > \tau_2$, which results in an investment of 0.5. For player two, we have $C_2 \leq \tau_2$ results an investment of 0.2. As a result, the total investment after two rounds is 1.2. Thus, player one obtains a payoff of $1 - 0.1 - 0.5 = 0.4$ and player two $1 - 0.5 - 0.2 = 0.3$. Players cannot spend more than what they have, we do not allow negative payoffs.

In one generation, many games with several rounds $r = 1, 2, ..., \Omega$ are played between groups of individuals chosen at random (herein we typically concentrate on the two-player game, but in methods we show that this is not a fundamental restriction of our model; see Methods section: Group Size). Each individual's final payoff, $\pi_i$, is then calculated as the average payoff of all games played by the individual in that generation. This payoff $\pi_i$ is translated into a fitness value $f_i = \exp[\pi_i]$[45]. Since payoff determines fitness, this means that the strategy that incurs the highest payoffs also has the highest probability of being transferred into the next generation.

We use a Wright–Fisher process to select the next generation of individuals where each offspring inherits the strategy of the parent (with the possibility of small errors). The individual's fitness is used to weigh the probability of an individual to contribute to the new population[20]. Errors in reproduction occur at the end of the generation with a probability $\mu$ for the thresholds $\tau$ and the contributions of each round independently. If they occur, errors in the threshold values add Gaussian noise with standard deviation $\sigma$ to them. Errors in the contributions are made using a uniform distribution between zero and $W$, see also[20].

Once a new population is produced, the process is repeated for multiple generations. Results reported in the figures represent the average of the dynamics over many generations once the random initial condition no longer affects the dynamics.

**Exploring the multi-loss game.** Initially, we explored the homogenous case where all players $i$ start with the same wealth $W_{i,0}$, the same risk probability $p$, and the same fraction of wealth lost $\alpha$. To model collective loss, we assume that in each round a fraction $\alpha$ of an individual's remaining wealth is lost with probability $p_r$.

We explore contributions for the case of linear risk curves (Fig. 5a) and their effects on contributions with respect to 1–4 round games (Figs. 1–5) and an 8 round game (Supplementary Fig. 2).

$$p(C_r) = 1 - \frac{C_r}{W_0}\lambda_1, \qquad (1)$$

where $W_0 = mW_{i,0}$ is the total wealth of the players at the start, $m$ is the number of players, $C_r$ is the total contribution in round $r$ and $\lambda_1$ controls how fast the risk declines.

We also use power function risk curves (Fig. 5b)

$$p(C_r) = 1 - \left(\frac{C_r}{W_0}\right)^{\lambda_2}, \qquad (2)$$

where $\lambda_2$ controls the shape of the risk curve (convex or concave). The third choice are step-like risk curves (Fig. 5c)

$$p(C_r) = \frac{1}{e^{\lambda_3\left(\frac{C_r}{W_0} - \frac{1}{2}\right)} + 1}, \qquad (3)$$

where the inflection point of the risk curve is at $\frac{1}{2}$ and $\lambda_3$ controls how rapid the risk declines.

These risk curves are normalized such that only the relative contribution $C_r/W_0$, i.e., the fraction of initial total wealth that is invested, enters.

**Group size.** While we concentrate on the case of $m = 2$ in the main text, our model allows for various group sizes. We find that large groups of such as $m = 8$ players typically contribute less than two-player groups in a single round game (Fig. 2 versus Supplementary Fig. 1). However, the multi-loss game quenches the group size effects (Supplementary Fig. 1a). Thus, just like the inclusion of time in general collective-risk, facilitates coordination[21], herein, we observe that the recurrence of loss can also increase cooperation.

**Heterogeneity.** Under wealth inequality, $W$ differs among players. For simplicity we only consider two types, rich and poor individuals, and explore the cases where $W_R \geq W_P$. We explore heterogeneity in the loss fraction $\alpha$, where individuals share the same risk probability, but differ in the proportion of their wealth they will lose, as shown in Fig. 3. We also incorporate heterogeneity in risk: where risk probability $p$ could differ between populations of rich and poor individuals, such that the total contributions results in different risk probability depend on the wealth status ($p_R[C_r] \neq p_P[C_r]$). The non-italic capital subscripts R and P now refer to rich and poor, while the italic subscript $r$ still stands for the current round.

**Risk heterogeneity.** We explore heterogeneity in risk: where risk probability $p$ could differ between populations of rich and poor individuals, such that the total contributions results in different risk probability depending on the wealth status ($p_R[C] \neq p_P[C]$ with the total contribution $C = c_R + c_P$), Supplementary Figure 3.

From now on we limit the results to one-round games ($\Omega = 1$) so that the $r$ subscript is no longer needed.

Using a general framework developed for the case of homogeneous wealth[34], we also analyze the expected contributions under wealth inequality and risk asymmetries for a one-round game. This method has broad applications, it allows us to understand the effect of general risk functions and their implications if we implement them into experimental setup. These experimental and theoretical insights could then be used to extrapolate how changes in perceived risk could affect climate change negotiations. This general framework can be applied to any risk function as has been previously demonstrated for a homogeneous situation[34]; however, herein, we broaden the analyses to heterogeneous situations that combines both wealth and risk inequalities. Since the heterogeneous setup of the game is inherently complex, we limit our analysis to the single round setup and consider a 2 player game, and ask how much would a focal player $f$ contribute if her endowment and her risk of losing it is greater or less than her co-player's.

The payoff of a player $f$ with endowment $W_f$ and who invests $c_f$ is given by

$$\pi_f = \left(W_f - c_f\right)\left(1 - p_f\right), \qquad (4)$$

where the probability of catastrophe is given by functions $p_f \in \{p_R, p_P\}$ for rich and poor players respectively, which depend on the total contribution $C = c_R + c_P$ and the total endowment $W = W_R + W_P$. We then can analyze under which circumstances these players cannot improve their payoffs from altering their behavior. In Supplementary Fig. 3 we do this numerically and then analyze the special case of linear risk curves analytically.

**Analytical approach**. While a full analytical treatment of our computational model is challenging due to finite population size and stochastic effects, we can improve our understanding by looking at the evolutionary stable strategies. As an example, we analyze the case of linearly declining risk more thoroughly. The rich have a higher endowment than the poor $W_R > W_P$ and a faster declining risk $p_R \le p_P$. While the risk for the poor approaches 0 only when both players contribute all their wealth ($C = W$), the risk for the rich is smaller and already 0 when $C = \frac{W}{\lambda_R}$ with $\lambda_R \ge 1$. Thus, we now set $\lambda_P = 1$ (in contrast to the figures above).

$$p_R = \begin{cases} 1 - \frac{C}{W}\lambda_R & \text{if } C < \frac{W}{\lambda_R} \\ 0 & \text{else} \end{cases} \qquad (5)$$

$$p_P = 1 - \frac{C}{W}. \qquad (6)$$

Then the average payoff of a player is

$$\pi_R = (W_R - c_R)(1 - p_R) = \begin{cases} (W_R - c_R)\frac{c_R + c_P}{W}\lambda_R & \text{if } c_R + c_P < \frac{W}{\lambda_R} \\ W_R - c_R & \text{else} \end{cases} \qquad (7)$$

$$\pi_P = (W_P - c_P)(1 - p_P) = (W_P - c_P)\frac{c_R + c_P}{W}. \qquad (8)$$

The evolutionary stable state is the payoff that decreases when the player deviates from his strategy (a local maximum). Since payoffs depend on the co-player's contribution we can only calculate the best response of a player to another players action.

In the case where the rich have a nonzero risk $\left(c_R + c_P < \frac{W}{\lambda_R}\right)$, we first take the derivative of the payoff to the player's contribution and set it to zero.

$$\frac{\partial \pi_R}{\partial c_R} = \lambda_R \frac{W_R - 2c_R - c_P}{W_R + W_P} \stackrel{!}{=} 0 \qquad (9)$$

$$\frac{\partial \pi_P}{\partial c_P} = \frac{W_P - 2c_P - c_R}{W_R + W_P} \stackrel{!}{=} 0 \qquad (10)$$

Then for $\lambda_R > 0$, $W_R + W_P \ne 0$ and $C = c_R + c_P < \frac{W}{\lambda_R}$ we get the best response functions in dependence of the co-player's contribution

$$c_R = \frac{1}{2}(W_R - c_P) \qquad (11)$$

$$c_P = \frac{1}{2}(W_P - c_R). \qquad (12)$$

In case the curves intersect in $0 \le c_R \le W_R$ and $0 \le c_P \le W_P$ and the intersection is in the region where the risk for the rich is still positive $C < \frac{W}{\lambda_R}$, we have the two best response functions (Eqs. (11) and (12)). The intersection gives the

**Table 1 Evolutionary stable state $c_P^*$ and $c_R^*$ for a decreasing linear risk curve for the rich $\lambda_R$ calculated by intersecting two linear curves**

| Assumption $W_R < 2W_P$ | $c_P^*$ | $c_R^*$ |
|---|---|---|
| $1 \le \lambda_R \le 3$ | $\frac{2W_P - W_R}{3}$ | $\frac{2W_R - W_P}{3}$ |
| $3 \le \lambda_R \le 2\frac{W_R + W_P}{W_P}$ | $W_P - \frac{W_R + W_P}{\lambda_R}$ | $2\frac{W_R + W_P}{\lambda_R} - W_P$ |
| $\lambda_R \ge 2\frac{W_R + W_P}{W_P}$ | $\frac{W_P}{2}$ | $0$ |
| Assumption $W_R > 2W_P$ | $c_P^*$ | $c_R^*$ |
| $1 \le \lambda_R \le 2\frac{W_R + W_P}{W_R}$ | $0$ | $\frac{W_R}{2}$ |
| $2\frac{W_R + W_P}{W_R} \le \lambda_R \le \frac{W_R + W_P}{W_P}$ | $0$ | $\frac{W_R + W_P}{\lambda_R}$ |
| $\frac{W_R + W_P}{W_P} \le \lambda_R \le 2\frac{W_R + W_P}{W_P}$ | $W_P - \frac{W_R + W_P}{\lambda_R}$ | $2\frac{W_R + W_P}{\lambda_R} - W_P$ |
| $2\frac{W_R + W_P}{W_P} \le \lambda_R$ | $\frac{W_P}{2}$ | $0$ |

evolutionary stable equilibrium

$$c_R^* = \frac{2W_R - W_P}{3} \qquad (13)$$

$$c_P^* = \frac{2W_P - W_R}{3}. \qquad (14)$$

We have repeatedly stated that this only holds for $C < \frac{W}{\lambda_R}$. But we also notice that for $c_P^* \ge 0$ we need $2W_P \ge W_R$. This means that the best response curves (Eqs. (11) and (12)) only cross to give the evolutionary stable state (Eqs. (13) and (14)) when the endowment of the rich is smaller than two times that of the poor. We thus divide our problem into two cases. Case 1 is when $W_R \le 2W_P$ and case 2 is when $W_R \ge 2W_P$, see Table 1.

Case 1: The above calculation assumes $C < \frac{W}{\lambda_R}$ but we have $C^* = c_R^* + c_P^* = \frac{W}{3}$ and so the above calculation is only valid for $\lambda_R \le 3$. For $\lambda_R > 3$ the risk for the rich vanishes $p_R = 0$ and so the payoff is maximized by using the minimal value of $c_R$ that keeps the risk at a value of 0 (Eq. (5)). Thus the best response for the rich is now $c_R = \frac{W}{\lambda_R} - c_P$ but the response for the poor stays the same (Eq. (12)). The intersection is

$$c_R^* = 2\frac{W_R + W_P}{\lambda_R} - W_P \qquad (15)$$

$$c_P^* = W_P - \frac{W_R + W_P}{\lambda_R}, \qquad (16)$$

which is non-negative for $\lambda_R < 2\frac{W_R + W_P}{W_P}$ (with $2\frac{W_R + W_P}{W_P} \in [4, 6]$ considering the condition for case 1. For even higher $\lambda_R$ the lines do not cross anymore. The minimum requirement for $p_R = 0$ is $c_R + c_P \ge \frac{W_P}{2}$ which corresponds exactly to the amount that the poor player contributes (Eq. (12)) if the rich contributes nothing, so $c_R^* = 0$ and $c_P^* = \frac{W_P}{2}$.

Case 2: When the endowment of the rich gets larger than two times the poor's ($W_R \ge 2W_P$) the two best response curves Eqs. (11) and (12) do not intersect anymore. Now there are four regimes of $\lambda_R$. If the poor contribute nothing, $c_P = 0$, the best response of the rich is to give half of their wealth $c_R^* = W_R/2$ (Eq. (11)). According to the condition $W_R \ge 2W_P$ the best response (Eq. (12)) for the poor is then $c_P = \left(W_P - c_R^*\right)/2 \le 0$. The poor player has no incentive to contribute and so the evolutionary stable state is $c_R^* = W_R/2$ and $c_P^* = 0$ as long as the condition $C < \frac{W}{\lambda_R}$ or (with $C = W_R/2$) $\lambda_R < 2\frac{W}{W_R}$ is fulfilled. As $\lambda_R$ increases to $\lambda_R \ge 2\frac{W}{W_R}$ the risk for the rich becomes 0 and we cannot calculate the best response. The rich now contribute as little as possible to just about fulfill this condition of zero risk ($c_R^* = \frac{W}{\lambda_R}$, see Eq. (5)) and the poor still pay nothing as long as $c_R^* > W_P$ (Eq. (12)). Once that threshold is met, which is equivalent to $\lambda_R \ge \frac{W}{W_P}$, the best response function for the poor (Eq. (12)) starts to matter and solving for this and the threshold condition (Eq. (5)) results in the optimal strategies $c_R^* = 2\frac{W}{\lambda_R} - W_P$ and $c_P^* = W_P - \frac{W}{\lambda_R}$. The value $c_R^*$ declines and $c_P^*$ increases as $\lambda_R$ increases further and when $c_P^* = \frac{W_P}{2}$ (equivalent to $\lambda_R = 2\frac{W}{W_P}$) the best response for the rich is again the minimum amount that results in $p_R = 0$, but since the poor already contribute enough this is no longer necessary $\left(c_R = \frac{W}{\lambda_R} - c_P^* \le 0\right)$ and thus $c_R^* = 0$. See Table 1 for a summary of the results. An alternative way to show the interplay between wealth and risk inequality is given in Supplementary Figure 4. With decreasing risk for the rich, they tend to invest less and at some point completely rely on the contributions of the poor.

**Code availability**. Our simulation code is available at https://github.com/abouchakra/Collective-Risk-Dilemma.

**Data availability**. The authors declare that no additional data was used in this study beyond that used in the code simulations.

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

## Author contributions

A.O., M.A.C., and A.T. developed the multiple loss model, M.A.C. and A.T. developed the rich and poor scenario. M.A.C., S.B., H.S., and A.T. analyzed the model. The final manuscript was written by all authors.

## Additional information

**Competing interests:** The authors declare no competing interests.

