## [Peer Review File · Nature Communications]

Reviewers' comments:

Reviewer #1 (Remarks to the Author):

Please find my report on the attached pdf.

[Editorial Note: This pdf is included on pages 2-8 of this report].

Report on “For uncertain climate change, immediate action is the best strategy”

I find this article very interesting but I do have some doubts. Let me start by making sure I understood the paper by writing a summary of my understanding of it. The authors should correct me if I missed something.

This work comprises the study of cooperative behavior for the individual management of public goods, here, applied to the problem of climate change mitigation.

The authors devise a repeated game based on the collective risk dilemma, a game that mimics the potential for disaster that collectively affects all players involved whenever contributions fall behind a given threshold. They introduce a generic risk function, p_r , which depends on the total contributions at any given round and sets the probability of disaster at that round. Disaster, in turn, is defined by a fixed fraction, α , of the player’s current wealth being lost, creating an effective (ensemble average) dampening of wealth of $1 - \alpha p_r$. In each round, players can decide to contribute to the game a given amount.

In this work, the authors chose to study the outcomes of 2-players games over a finite set of R rounds. The set of possible strategies consists of R thresholds, τ_r , and two contribution levels, a and b : in each round, if the total contributions so far, C_r , is above the threshold of a player, $C_r > \tau_r$, he contributes a , else, $C_r < \tau_r$, he contributes b . The success of a (player with a given) strategy, i.e. the success of the set $(\{\tau_r\}; a, b)$, is obtained by computing the average payoff of that strategy when playing against several different strategies in a population and setting a probability of propagation of the strategy in the population that is a growing function of that payoff. The most successful strategies, in this sense, spread in the population.

The authors study this model through simulations. They start by looking at the effect the different risk functions, $p_r \equiv p(C_r)$, and loss factor α , which effectively rescales p_r – have on the average contributions and they explore this in games with different durations, $R = 1, 2$, and 4 . They show that α increases cooperation but, for longer games, in this case $R = 4$, intermediate values of α can maximize the average contribution; though not for all shapes of p . In any scenario, a non-zero risk of disaster induces cooperation – which corroborates previous results.

On a second stage, the authors extend the functional form of p_r taking an explicit dependence on the round. It is not very clear to me, but I assume there is still a dependence in the total contributions but in this case the author redefined p_r such that $p_r = p(C_r)d_r$, where d_r is some binary function that determines if disaster may actually happen in that round or not. So, in the first results, scenario i), disaster could happen every round, $d_r = 1$, and at this stage it can happen in three other scenarios: ii) only on first round, $d_r = \delta_{0r}$, iii) only on last round, $d_r = \delta_{Rr}$, and iv) a random round, $X \sim \text{Unifom}(1, R)$, setting $d_r = \delta_{rX}$. The authors find similarities between scenario i) and ii) and between iii) and iv). Most importantly, they find a common strategy of increased contributions in the initial rounds to be the most successful.

I find the “multiple loss model” a very valuable improvement in the approach to the study of risky games with long duration, being important to scientists in the field of Evolutionary Game Theory, and also experimental economics and climate and other public goods

management. As I see it, the rounds conceptually map into real world time, whereas the evolution (or Evolutionary Game Theory) is used as a tool to find prevalence of strategies under frequency dependent scenarios. The most notable result relates to the last figure (Figure 3) in which the authors show that two different successful strategies are possible: the “wait and see” and the “act quickly” strategies. Nonetheless, the authors could improve the manuscript by making it clearer if they were able to show that one is more prevalent than the other or why is that so.

Let me start with a minor point, which relates exactly to figure 3, that then builds into a more general concern. Given that the whole analysis of the figure is based on the contributions per round, wouldn't the results be much clearer if the axis were Rounds-Contributions (x-y)? We could then see in which round contributions are larger and the authors could even signal where the disaster happens. In my mind, this makes sense since given that, aside from the name, there is not really a relationship between the different lambdas and I failed to get an intuitive interpretation for it, so it becomes just a parameter to check the generality of the results; and that could be represented by different lines or a shade with some standard deviation for the ensemble of parameters. Additionally, one should then ask what happens for the remaining parameters. As far as I understand, there is also W_0 , σ , and μ . Ideally, σ and μ should just help getting to equilibrium faster but W_0 does bring the different effective (average payoffs) functions closer to each other: take for instance the step function for small λ , that is basically a straight line at $1/2$, and the linear risk curve for small lambda, that is a straight line near 1. Setting larger W_0 for the step function may bring the two together and the results that seemed to not happen for the threshold function then show up but only for larger W_0 .

My second point is related exactly with the strategies “wait and see” and “act quickly”. In fact, the players are using strategies $(\{\tau_r\}; a, b)$, and those are the ones under selection. Can we understand the average levels of cooperation in the light of the most frequent strategy? What is a “wait and see” strategy and a “act quickly” strategy in terms of the $(\{\tau_r\}; a, b)$? Is this clear from the results of the simulations or just from the average outcome? It seems peculiar that by the end of the article, we don't know what strategies the individuals are actually adopting. Are they too diverse to reach any conclusion? Is $a > b$, generally, or vice-versa? What is the average threshold per round? Also, are the results driven by noise, in the sense that it requires heterogeneous populations to get the two strategies the authors find?

A more abstract comment has to do with a measure of success in the different scenarios. Since the authors are talking about climate change, and the ultimate goal is to understand how well the different strategies are doing in terms of preserving the public good, I believe a measure of this would give the article a bigger statement in terms of the effects of different macro-strategies (let me call them that way, to distinguish from the strategies the individuals actually play). Indeed, we know that higher total contributions do not mean better outcomes: if the contributions come too late, they might not solve any problem and, additionally, the lack of initial contributions may have caused too many disasters on the way. I could think of average number of disasters as a good measure to explore.

As a third point, I have to say that I am concerned with the one equation of the paper. See: The authors say that, in a given round, if a loss happens, the current endowment is lost. However, when we look at equation [1] a loss in the first round does not affect the Endowment that is transported to the second round, it is as if the two disasters affect, at the same time, the last round. Is this just some mistake in the formula and the authors did not use the formula for the simulations or is this error propagated?

Besides correcting the formula, it would be interesting to see some partially analytical treatment of this framework/game. See my notes below, which contain quite a bit of hand-waving and back of the envelope calculations. Feel free to use them, as I think the paper would become much clearer from the game theoretical point of view: it would allow one to understand what is going on in terms of the interplay between the shape of the risk function and the effective costs and benefits of this iterative game.

My last three points are more suggestions than really things that need to be deeply incorporated in the manuscript. i) Why is the whole analysis limited to 4 rounds? Is it due to computational limitations? In the limit, one could think of a continuous time that describes the real decision process under an evolutionary-time that provides evolutionary stable distributions of strategies. ii) Is the scenario of a disaster happening in the first round different from a game with a single round? If so, why? iii) It would be nice to see scenarios in which one maps the perception of closeness (in time) of disaster by setting a threshold value above which disaster actually happens. I.e., studying the scenarios in which the disaster occurs every round, only after one round, after two rounds, after three rounds, ..., after M rounds. In the previous notation would be something like setting $d_r = \Theta(r - M)$. That would give a clear portrait of the effect of discounting the proximity of disaster from a (evolutionary) game theoretical point of view.

Overall, as far as I am aware, the paper brings novelty to the area, thus being of interest to the experts on the field. Furthermore, it will provide a new setup to explore the outcome of this dilemma we so deeply need to understand. Once the authors address extended exploration of parameters, analytical intuition on the output, and clarification of the interpretation of the model, if the results still hold (which I believe to be the case), the claims will be stronger and sound. The manuscript is well written but is lacking simplification of arguments and missing conclusions that follow more directly from the results shown (or maybe missing the results that lead to those conclusions). This would create a shorter version of the present paper, opening space for the additional materials necessary. This said, and being clear that I have a positive opinion of the paper, I believe the authors should be allowed to revise their manuscript to address the specific concerns before a final decision is reached.

Below the authors can find some notes written as I read through the paper as well as the notes on the average payoff structure.

Best regards

Vitor V. Vasconcelos

In here, I go through the paper and leave the authors some other questions and comments that might help improve the manuscript. Even if some things seem irrelevant, I believe clarification of these small questions would make the manuscript more sound both scientifically and literarily.

Abstract:

Line 12: *devise a game where the risk of a collective loss is unpredictable or can reoccur.*
Or? Or do you mean *and*?

“wait and see” and “act quickly” strategy is not defined in the abstract nor it is said that will be defined in the article. Maybe drop strategy, if the authors mean exactly what the usual words mean.

Line 12: *when risk declines rapidly and its timing is known.*
I am not sure what “timing of risk” means. Could you clarify?

Line 15: *we find that catastrophic scenarios are not necessary to induce such immediate collective action. In most scenarios...*

Even after reading the paper I don't really know which ones are the catastrophic scenarios and which ones aren't. Maybe I missed it, but it was only in the abstract and in the conclusions that I found this distinction between mild and catastrophic scenarios. It would help the manuscript clarifying which ones are which in the discussion.

Also notice that the careful reader of the abstract sees the scenarios unpredictable loss vs reoccurring loss and different declining risks and known vs unknown timing and all of them seem catastrophic.

Introduction:

The paragraph before line 42

Explicitly stating that the different real world examples mentioned are expectedly asynchronous and thus represent reoccurrences of disasters with the same fundamental origin would clarify the importance of the manuscript, besides creating a bridge with the following paragraph.

Line 48: *individuals do not know when the risk will hit.*
Is it the risk that hits? Or the disaster.

Results:

Line 71: *We apply evolutionary game theory to understand and identify the set of stable contributions under various risk scenarios.*

Do the authors agree that *stable* here is being used in a very loose sense? In this case EGT does not provide stable contributions but a dynamically stable distribution of strategies that comprise different thresholds and contributions. The average contributions per round may be an emergent pattern, though... They could also be stable to changes in parameters or simply convergent in time.

Line 73: *Stability means that that a player would have no incentive to change her contribution*

This follows in line with my previous comment about EGT, what the outcome is and what the meaning of *stable* two sentences before is. The authors seem to be describing concepts closer to Nash and other equilibria that assume full knowledge of the game. As the authors later clarify, this is not the case with EGT.

Line 86: Intermediate values of α lead to differentiated contributions because of the complex interactions between the fraction of wealth lost and the risk curve.

What do you mean by differentiated contributions? Are they differentiated by amounts, rounds, players, ...?

Line 89: The reason is that intermediate potential losses imply an inherent uncertainty as the number of rounds increase —intermediate losses may accumulate over the course of the game, adding up to a large overall loss.

The reasoning for optimal alpha seems obscure and probably follows from the intuition the authors derive from the study they did but does not follow through the results the readers see. Perhaps the analytical treatment would clarify this.

In figure 1

Some scenarios are much more risky than others as they systematically present higher probabilities of failure. Is there a way to compare scenarios that are similar? Is this how the authors classify mild and catastrophic scenarios?

Line 108: To summarize, we find that for the game with multiple losses, high initial risks and full losses are not necessary to observe cooperative

Isn't this written two sentences before?

Figure 3:

Again, either changing the x-axis to round or explicitly write round 1, round 2, ... in the different colors would make the figure more readable. For a while I thought $r=1$ meant only one round, and so on. Clearly, nothing is missing and with some effort one can interpret the figure. But all the arguments are of contributions over time and that is not what comes immediately from the image.

Line 118: the game with multiple losses elicits a similar amount of contributions as the game where the potential loss only occurs in the first round

Not for the threshold game with those parameters. The authors could clarify what they mean by similar. There are similarities and differences between all the scenarios. It would be important to clarify what those similarities are.

If the authors are comparing total contribution, it would be good to have a cumulative contribution line to really compare the scenarios in what total contribution is concerned.

Line: 123: It seems that if risk is in a fixed round and exhibits strong threshold effects, individuals can time their contributions accordingly. However moving beyond this particular case, we find that an “act quickly” strategy is most robust.

It seems that according to what?

Conclusions:

Line 131: how rapidly losses increase with the intensity of the events

Aren't losses and intensity of an event the same thing over all the models that we see in the literature? As I see it, one assumes losses as a fundamental quantity in the decision making, i.e., the intensity of an event is the losses it causes to a single individual. Could the authors clarify?

Line 156: We have shown that under uncertainty in terms of the timing of losses, individuals increase their efforts to ensure contributions are sufficient not only for now, but that their future is also protected. Importantly, our results provide another compelling argument for immediate climate action.

I find this argument a little confusing. The model is not descriptive of the actions of individuals, right? The model finds from an EGT point of view the strategies that are most fit to deal with the game that is posed to the players. Otherwise, one could ask "why most nations do not contribute to climate mitigation?". The last sentence, on the other hand, does make sense in the light of the interpretation that the model is capturing the best unilateral responses if we agree the game is the one being played by the actors. I feel the manuscript would have a lot to gain if this argument is clear from the beginning and one leaves no doubt of what the model is trying to capture: is it describing the behavior of players over time when they face this dilemma or is it finding the patterns of contributions that are able to persist when players are adopting the most successful behaviors in a rationally bounded environment?

Methods:

Line 162: We use evolutionary game theory to identify robust contributions between interacting players

Robust over time, you mean? Or are we talking about "evolutionary robust strategies"?

Line 163: This means players do not know the structure of the game and cannot apply advanced reasoning about their possible actions.

Exactly, which seems to contradict the definition of stability in line 73.

170: [Is $a > b$ or vice versa? What is the strategy that emerges?]

Line 169: that depends on the collective contributions accumulated over all rounds so far, C_r

The threshold depends on the contributions? Or on the round? I thought τ_r was a set of numbers that defined the strategy of a player. Is this correct?

Line 179: Errors occur with a probability p for the thresholds τ and the contributions of each round independently. If they occur, errors in the threshold values add Gaussian noise with standard deviation σ to them. Errors in the contributions are made using a uniform distribution.

It would improve reproducibility if the authors clarify if errors occur in the reproduction process or during the game. Also, the uniform distribution is between what and what? 0 and W_0 ? Can players end the rounds with negative endowment?

Other: There is a lost full stop in line 167.

Notes on average Payoff

The payoff can be computed as the expected endowment in the last round. If you let $\{d_r\} = \{d_1, \dots, d_R\}$ be a binary vector in which 1's represent the occurrence of a disaster, then

$$P(\{d_r\}) = \prod_{r=1}^R (1 - p_r)^{1-d_r} p_r^{d_r} .$$

With this, one can compute

$$\Pi_i = \sum_{\{d_r\}} P(\{d_r\}) \left(\prod_{r=1}^R (1 - \alpha)^{d_r} W_0 - \sum_{r=1}^R c_{ri} \prod_{k=r}^R (1 - \alpha)^{d_k} \right)$$

Which can be written as

$$\Pi_i = W_0 \prod_{r=1}^R (1 - \alpha p_r) - \sum_{r=1}^R c_{ri} \prod_{k=r}^R (1 - \alpha p_k) \quad 1$$

Which, for computational purposes, can be given by

given $X = W_{i0}$, do $X \leftarrow (1 - p_r \alpha)(X - c_{ir})$ with $i = 1, \dots, R$.

This is fairly easy to interpret as it means that in each round, on average, the endowment is reduce by a factor of $p_r \alpha$, which is what I meant when I described the effect of α in the first paragraphs.

The particular case of 2 rounds becomes

$$\Pi_i = (1 - p_1 \alpha) \left((1 - p_1 \alpha)(W_0 - c_{i1}) - c_{i2} \right).$$

First, this proofs that α and p_r contribute equally to the process and justifies fixing a maximum of 1 for p_r , since α is able to rescale that maximum from 0 to 1. Additionally, one can look at the marginal gains of switching to some other contribution (by whichever mechanism: changing a , changing b , changing τ) in any given round, which, ultimately, is what drives the replication process. If a given player changes c_{iK} to $c_{iK} + \delta$, in round K , then one could look at a small δ and show that increasing a contribution will increase the first term of Eq.(1) (effective initial endowment W_0) by an amount that depends on the derivative of $p(C_r)$ but will also increase the second term (effective cost) by an amount that is higher the later on round the contribution happens, thus explaining the “quick action” result.

Reviewer #2 (Remarks to the Author):

Our understanding of the underlying mechanisms responsible for promoting and maintaining cooperation within the context of a collective-risk social dilemma (applicable to the prevention of climate change scenarios) is limited to some extent, despite many excellent papers appearing in the related literature in recent years.

This paper continues this line of work by presenting an analysis of an extension to the collective-risk social dilemma, using evolutionary game theory. As such, the paper represents a small incremental study, extending earlier work from some of the authors (eg., [18] [19]), rather than a describing novel research direction.

The authors explicitly examine the perception of risk, and associated implications when making a decision (contribution), by focussing on specific risk curves and timing of 'disturbances' (of varying severity?) in the game. They develop a game where individuals are faced with the risk of a loss in a specific round of the game, which can be a 'one-off' single round occurrence or could occur in random round of the game. Exploring the effects of a potential losses in this setting is interesting, especially when individuals can retain some of the resource after an 'event.' This approach is reasonable given the constraints of the adopted abstract modelling framework.

The paper is generally well written. However, some aspects of the approach/methodology require clarification. For example, further details describing the definition of (distinction between) 'games per generation and the number of rounds in a game would be of benefit to the reader, allowing other researchers to implement the model unambiguously. Also, further clarification for the selection of risk curves should be included (was it based solely on previous work?).

Detailed simulation results and analysis have been included in the paper. The metrics used for evaluation were defined clearly. The results reported in Fig 2-3 are 'steady-state' values (?). Representative temporal dynamics could also be included, illustrating the effectiveness of the learning process (particularly in terms of risk curves and the 'severity of any loss.' Additional discussion of the effects of parameter values (as depicted in Fig 3) is warranted. Also, there is room to explain/justify the appropriate similarity between any risk curve for large λ . More generally, why does a 'power-function lead to results that appear to be significantly different to 'step-like' functions across λ values? (see Fig 3a Fig 3b)

On the whole, the results and analysis are appropriate, supporting the claims made in the text. The conclusion is appropriate, and has been discussed in the wider context of coupled human-social-ecological systems.

Additional minor comments:

- * Does stability within the model (see lines 73-75) always emerge (under all circumstances)?
- * What impact does the shape of the risk curve actually have? Can a clear relationship in terms of λ be stated?
- * What is the analytical prediction for the game outcome over multiple rounds? How does this depend on the risk curve/parameter values?

* System dynamics are guided by the embedded 'social learning' /evolutionary updated mechanism. Further analysis/description would be interesting.

* How does the unpredictable nature of a 'catastrophic disturbance' impact system dynamics over (a) longer time periods/rounds; (b) larger population sizes; and (c) the value of α for these different parameter values?

* There is room to clarify the captions in Fig 1 and 2

* Fig 3 suggests that there is significant difference in the performance metric (av contribution) for specific risk curves, λ values, with magnifications of differences in the early round. The summary of results is consistent with the plots, however, what happens over longer time periods / rounds, especially for larger λ values? For small λ values, I am not convinced that a general conclusion can be made (based on the evidence presented). Further theoretical analysis and supporting numerical simulations would help to clarify these points.

* From line 117 ... "Interestingly, the game with multiple losses elicits a similar amount of contributions as the game where the potential loss only occurs in the first round" ... Further discussion/clarification would enhance the paper.

* From line 149-150 "This is a striking difference between our current and previous research on climate cooperation".... Perhaps you could highlight the fact that the model described in this paper represents a small incremental step.

Reviewer #3 (Remarks to the Author):

Nature Communications Review: "For uncertain climate change, immediate action is the best strategy"

The authors use evolutionary game theory to model a collective-risk game wherein individuals may suffer losses of varying degrees over one to four rounds of play. The authors examine the effects of the timing, probability, and extent of damages on average contributions using simulations of a model developed specifically to look at the question of whether delayed losses affect the contribution profiles of individuals in the game.

The paper touches on a few important aspects of climate change that have not been sufficiently covered in the (experimental) literature on international climate cooperation (e.g., Milinski et al., 2008; Milinski et al., 2011; Tavoni et al., 2011; Dannenberg et al., 2014). For instance, the uncertainty of the timing of losses and the possibility of recurring losses are both important features of the climate problem that require further scrutiny to enhance the generalizability of the existing literature (which is more or less confined to theoretical and experimental investigation due to the difficulty of implementing field studies on the subject). Additionally, the cumulative effects of contributions toward reducing future risk represent a realistic component of the climate problem often neglected in the literature.

While the authors appear to situate the paper within the experimental literature on international climate cooperation (see reference to Milinski et al., 2011; line 43), the study presented here uses evolutionary game theory to draw conclusions regarding individuals' responses to various features of climate damages. However, these models may produce questionable predictions, as evidenced through comparison with observed outcomes. For example, Smead et al.'s (2014) finding that heterogeneity of endowments (wealth) actually increases cooperation goes against both intuition and all theoretical and experimental results of which I am aware (e.g., Barrett, 2004; Tavoni et al., 2011), and other conclusions drawn in the Smead paper (e.g., regarding restricting private demands) were not replicable in a lab experiment designed to closely model their ABM (Gosnell and Tavoni, 2017). Therefore, perhaps aside from its specific application here, I am rather skeptical of the method's ability to predict actual behavior amidst convoluted incentive structures outlined in the behavioral economics and political economy literatures.

Moreover, the usefulness of the main findings themselves may be limited. The authors claim that the "act quickly strategy is most robust due to uncertainty", which simply endorses adoption of the precautionary principle in a situation where the necessity of early action is widely accepted. The existing question is not one of whether acting quickly is the correct strategy in the presence of multiple dimensions of uncertainty; instead, it is how to successfully promote and achieve such near-term cooperation (i.e. it is a political economy / human behavior problem). What happens when we play this game with actual people, and with (many) heterogeneous players?

It seems to me this paper could be substantially enhanced if the authors complement the model with a behavioral experiment that allows for direct comparison of the results from their simulations to observed outcomes from complex and non-deterministic human interactions; the outcomes of both methods may then be compared and contrasted with established economic theories and various relevant real-world contexts (especially international climate negotiations) to comment on generalizability. The experimental addition would not only allow the research to avoid critique regarding the general oversight of AB models' explanatory performance (e.g., see Windrum et al., 2007), it would also provide methodological insight and a means for direct comparison of the outcomes of the model with the experimental literature cited in the paper's introduction. Furthermore, for the sake of replicability and transparency, it would be a good idea for the authors to make their source code available with the publication, and perhaps also provide (supplementary) information with regards to the model's sensitivity to the chosen parameters.

Minor edits:

- "Reoccur" and "reoccurrence" should be replaced with "recur" and "recurrence" throughout the manuscript
- First sentence of abstract: Climate change will not be mitigated; perhaps change to "dangerous climate change"? (And "requires" global cooperation.)
 - o First sentence of paper: is it true that "everyone" needs to exert effort to reduce climate change effects? Consider rephrasing this sentence.
- Change the comma after "debatable" to a semicolon.
- Line 23: decide "whether" to contribute (remove "or not").
- Lines 24-25: "But in light of dangerous climate change..." Why? Due to the framing of losses/damages rather than benefits accrued from the public good? Justify this claim.
- Lines 39-40: commas after "Also" and after "will be"
- Lines 42-44: this sentence needs some restructuring or grammatical editing.
- Lines 51-97: These are not results nor discussion, this is your model; perhaps create a section for this part that precedes the results section.
- Line 114 (and subsequent text): How is the overall probability of losses held constant when losses occur either one time or multiple times? This was not clear to me from the text in the manuscript, and further explanation may be merited.
- Line 167: remove the full stop after 'strategy' (and add one to line 184 after 'dynamics').

Reviewers' comments:

Reviewer #1 (Remarks to the Author):

Report on “For uncertain climate change, immediate action is the best strategy”
I find this article very interesting but I do have some doubts. Let me start by making sure I understood the paper by writing a summary of my understanding of it. The authors should correct me if I missed something. This work comprises the study of cooperative behavior for the individual management of public goods, here, applied to the problem of climate change mitigation.

The authors devise a repeated game based on the collective risk dilemma, a game that mimics the potential for disaster that collectively affects all players involved whenever contributions fall behind a given threshold. They introduce a generic risk function, p_r , which depends on the total contributions at any given round and sets the probability of disaster at that round. Disaster, in turn, is defined by a fixed fraction, α , of the player's current wealth being lost, creating an effective (ensemble average) dampening of wealth of $1 - \alpha p_r$. In each round, players can decide to contribute to the game a given amount.

In this work, the authors chose to study the outcomes of 2-players games over a finite set of R rounds. The set of possible strategies consists of R thresholds, τ_r , and two contribution levels, a and b : in each round, if the total contributions so far, C_r , is above the threshold of a player, $C_r > \tau_r$, he contributes a , else, $C_r < \tau_r$, he contributes b . The success of a (player with a given) strategy, i.e. the success of the set $(\{\tau_r\}; a, b)$, is obtained by computing the average payoff of that strategy when playing against several different strategies in a population and setting a probability of propagation of the strategy in the population that is a growing function of that payoff. The most successful strategies, in this sense, spread in the population.

The authors study this model through simulations. They start by looking at the effect the different risk functions, $p_r \rightarrow p(C_r)$, and loss factor $-\alpha$, which effectively rescales p_r – have on the average contributions and they explore this in games with different durations, $R=1, 2$, and 4. They show that α increases cooperation but, for longer games, in this case $R=4$, intermediate values of α can maximize the average contribution; though not for all shapes of p . In any scenario, a non-zero risk of disaster induces cooperation – which corroborates previous results. On a second stage, the authors extend the functional form of p_r taking an explicit dependence on the round. It is not very clear to me, but I assume there is still a dependence in the total contributions but in this case the author redefined p_r such that $p_r = p(C_r) d_r$, where d_r is some binary function that determines if disaster may actually happen in that round or not. So, in the first results, scenario i), disaster could happen every round, $d_r=1$, and at this stage it can happen in

three other scenarios: ii) only on first round, $dr = \delta 0r$, iii) only on last round, $dr = \delta Rr$, and iv) a random round, $X \sim \text{Unifom}(1, R)$, setting $dr = \delta rX$. The authors find similarities between scenario i) and ii) and between iii) and iv). Most importantly, they find a common strategy of increased contributions in the initial rounds to be the most successful.

I find the “multiple loss model” a very valuable improvement in the approach to the study of risky games with long duration, being important to scientists in the field of Evolutionary Game Theory, and also experimental economics and climate and other public goods management. As I see it, the rounds conceptually map into real world time, whereas the evolution (or Evolutionary Game Theory) is used as a tool to find prevalence of strategies under frequency dependent scenarios. The most notable result relates to the last figure (Figure 3) in which the authors show that two different successful strategies are possible: the “wait and see” and the “act quickly” strategies.

Thank you for the description of our model and for the positive and encouraging comments. Technically, our game is not repeated – there are R rounds, but the resulting payoff contributes to fitness only once. The effects that are usually considered in repeated games are not present in our model.

Nonetheless, the authors could improve the manuscript by making it clearer if they were able to show that one is more prevalent than the other or why is that so.

Let me start with a minor point, which relates exactly to figure 3, that then builds into a more general concern. Given that the whole analysis of the figure is based on the contributions per round, wouldn't the results be much clearer if the axis were Rounds-Contributions (x - y)? We could then see in which round contributions are larger and the authors could even signal where the disaster happens. In my mind, this makes sense since given that, aside from the name, there is not really a relationship between the different lambdas and I failed to get an intuitive interpretation for it, so it becomes just a parameter to check the generality of the results; and that could be represented by different lines or a shade with some standard deviation for the ensemble of parameters.

Additionally, one should then ask what happens for the remaining parameters. As far as I understand, there is also $W0$, σ , and μ . Ideally, σ and μ should just help getting to equilibrium faster but $W0$ does bring the different effective (average payoffs) functions closer to each other: take for instance the step function for small λ , that is basically a straight line at $\frac{1}{2}$, and the linear risk curve for small lambda, that is a straight line near 1. Setting larger $W0$ for the step function may bring the two together and the results that seemed to not happen for the threshold function then show up but only for larger $W0$.

We agree, now we have added a new figure 3 where we show round vs contribution. For the remaining parameters we have added the previous figure to the supplementary, Fig S2. We agree that we have focused too much on the different curves in the previous version and also renamed the lambda parameters to distinguish them.

My second point is related exactly with the strategies “wait and see” and “act quickly”. In fact, the players are using strategies $(\{\tau r\}; a, b)$, and those are the ones under selection. Can we understand the average levels of cooperation in the light of the most frequent strategy? What is a “wait and see” strategy and a “act quickly” strategy in terms of the $(\{\tau r\}; a, b)$? Is this clear from the results of the simulations or just from the average outcome? It seems peculiar that by the end of the article, we don't know what strategies the individuals are actually adopting. Are they too diverse to reach any conclusion? Is $a > b$, generally, or vice-versa? What is the average threshold per round? Also, are the results driven by noise, in the sense that it requires heterogeneous populations to get the two strategies the authors find?

In the continuous case of strategy space it is almost impossible to narrow down to the exact strategy (corresponding to a “genotype”), because there are many ways in which a player can contribute the same or a similar amount (corresponding to a “phenotype”). Our simulation average only shows the actions of players -- we did not attempt to search within the strategy space, as it seems of minor interest how exactly these behaviors are encoded. However, we have explored this exact question in a discontinuous action space in a previous study. We now have clarified this issue by describing the outcomes as behaviors and not strategies.

A more abstract comment has to do with a measure of success in the different scenarios. Since the authors are talking about climate change, and the ultimate goal is to understand how well the different strategies are doing in terms of preserving the public good, I believe a measure of this would give the article a bigger statement in terms of the effects of different macro-strategies (let me call them that way, to distinguish from the strategies the individuals actually play). Indeed, we know that higher total contributions do not mean better outcomes: if the contributions come too late, they might not solve any problem and, additionally, the lack of initial contributions may have caused too many disasters on the way. I could think of average number of disasters as a good measure to explore.

Thank you, we agree that our measure of success may not be intuitive. We have now included in Fig 3 how the player's contributions reduce the risk, which is an alternative measure of the average number of disasters. The reduction of risk translates into protecting the good in climate change. But importantly, in our case strategies that maximize personal success spread and success of the group in reducing the number of disasters is just a by-product

As a third point, I have to say that I am concerned with the one equation of the paper. See:

The authors say that, in a given round, if a loss happens, the current endowment is lost. However, when we look at equation [1] a loss in the first round does not affect the Endowment that is transported to the second round, it is as if the two disasters affect, at the same time, the last round. Is this just some mistake in the formula and the authors did not use the formula for the simulations or is this error propagated?

Thank you very much for capturing this crucial point. This is an embarrassing oversight on our part in terms of describing the payoffs arising from an earlier version of the manuscript. You are right, the remaining endowment of course has to reflect previous losses. We have now corrected the equation, but we decided to not overload our notation by the additional condition that contributions must be below the current endowment (which is important for large α). Fortunately, this does not have any impact on our other results and simulations since they are not based on this equation (and the equation is only there for illustrative purposes).

Besides correcting the formula, it would be interesting to see some partially analytical treatment of this framework/game. See my notes below, which contain quite a bit of hand-waving and back of the envelope calculations. Feel free to use them, as I think the paper would become much clearer from the game theoretical point of view: it would allow one to understand what is going on in terms of the interplay between the shape of the risk function and the effective costs and benefits of this iterative game.

Thank you. We have a general frame work for a single round game that was previously developed for the original game (Hagel et al., Scientific Reports 2016). However, this game grows in complexity quickly such that an analytical calculation would only work in specific circumstances and not as a general approach. Please refer to the new supplementary section called "analytical results".

My last three points are more suggestions than really things that need to be

deeply incorporated in the manuscript. i) Why is the whole analysis limited to 4 rounds? Is it due to computational limitations? In the limit, one could think of a continuous time that describes the real decision process under an evolutionary-time that provides evolutionary stable distributions of strategies.

Our model is not limited by computational considerations, we wanted to maintain a balance with simplicity and complexity of the model. We have previously explored a related game with up to 20 rounds. But since now we include a continuous strategy space, it seems to be redundant to explore a large number of rounds with such a large parameter space – in particular if the trends are similar. Moreover, the setup is already so complex that it does not seem to be easy to grasp it.

ii) Is the scenario of a disaster happening in the first round different from a game with a single round? If so, why?

No, as you anticipate they are identical. This is shown in our new Figure 3.

iii) It would be nice to see scenarios in which one maps the perception of closeness (in time) of disaster by setting a threshold value above which disaster actually happens. I.e., studying the scenarios in which the disaster occurs every round, only after one round, after two rounds, after three rounds, ..., after M rounds. In the previous notation would be something like setting $dr = \Theta(r - M)$. That would give a clear portrait of the effect of discounting the proximity of disaster from a (evolutionary) game theoretical point of view.

Overall, as far as I am aware, the paper brings novelty to the area, thus being of interest to the experts on the field. Furthermore, it will provide a new setup to explore the outcome of this dilemma we so deeply need to understand. Once the authors address extended exploration of parameters, analytical intuition on the output, and clarification of the interpretation of the model, if the results still hold (which I believe to be the case), the claims will be stronger and sound.

Thank you for the kind and positive comments, we hope our revised version is clearer. We have now extended the model in several ways and added more of the extensive parameter exploration that we have performed before.

The manuscript is well written but is lacking simplification of arguments and missing conclusions that follow more directly from the results shown (or maybe missing the results that lead to those conclusions). This would create a shorter version of the present paper, opening space for the additional materials

necessary. This said, and being clear that I have a positive opinion of the paper, I believe the authors should be allowed to revise their manuscript to address the specific concerns before a final decision is reached.

Thank you for the kind and positive comments. We have now re-organized the paper, moved the exploration of risk curve parameters to the supplement and added additional material that should in our opinion make it easier to understand the gist of the model.

Below the authors can find some notes written as I read through the paper as well as the notes on the average payoff structure.

In here, I go through the paper and leave the authors some other questions and comments that might help improve the manuscript. Even if some things seem irrelevant, I believe clarification of these small questions would make the manuscript more sound both scientifically and literarily.

Abstract:

Line 12: devise a game where the risk of a collective loss is unpredictable or can reoccur.

Or? Or do you mean and? “wait and see” and “act quickly” strategy is not defined in the abstract nor it is said that will be defined in the article. Maybe drop strategy, if the authors mean exactly what the usual words mean.

Thank you. Changed.

Line 12: when risk declines rapidly and its timing is known.
I am not sure what “timing of risk” means. Could you clarify?

Thank you, we have clarified these statements now. We referred to situations where the loss can occur in a random round.

Line 15: we find that catastrophic scenarios are not necessary to induce such immediate collective action. In most scenarios...

Even after reading the paper I don't really know which ones are the catastrophic scenarios and which ones aren't. Maybe I missed it, but it was only in the abstract and in the conclusions that I found this distinction between mild and catastrophic scenarios. It would help the manuscript clarifying which ones are which in the discussion.

Also notice that the careful reader of the abstract sees the scenarios unpredictable loss vs reoccurring loss and different declining risks and known vs

unknown timing and all of them seem catastrophic.

Thank you, we have clarified these statements. Indeed, there is not a clear distinction anymore, as the fraction of wealth lost can range from 0 to 1. We referred to the case of $\alpha=1$ here.

Introduction:

The paragraph before line 42

Explicitly stating that the different real world examples mentioned are expectedly asynchronous and thus represent reoccurrences of disasters with the same fundamental origin would clarify the importance of the manuscript, besides creating a bridge with the following paragraph.

Thank you, we have added a sentence to clarify this issue.

Line 48: individuals do not know when the risk will hit.

Is it the risk that hits? Or the disaster.

Thank you, we have corrected the sentence.

Results:

Line 71: We apply evolutionary game theory to understand and identify the set of stable contributions under various risk scenarios.

Do the authors agree that stable here is being used in a very loose sense? In this case EGT does not provide stable contributions but a dynamically stable distribution of strategies that comprise different thresholds and contributions. The average contributions per round may be an emergent pattern, though... They could also be stable to changes in parameters or simply convergent in time.

We agree that we use the term “stable” in a loose sense. For the evolutionary algorithm we use, the connection to stability in a classical game theoretical sense is not yet formally established. We also agree that the underlying coding of behaviors leaves room for variation, as only the contributions determine fitness. However, since evolutionary models typically impose higher requirements on stability than classical game theory, we believe that our analysis is meaningful.

Line 73: Stability means that that a player would have no incentive to change her contribution This follows in line with my previous comment about EGT, what the outcome is and what the meaning of stable two sentences before is. The authors seem to be describing concepts closer to Nash and other equilibria that assume full knowledge of the game. As the authors later clarify, this is not the case with

EGT.

We agree, but since any ESS is also a Nash equilibrium, there is an intimate connection between the two. However, we agree that our explanation for stability did not resonate with our evolutionary approach and thus we have changed that sentence.

Line 86: “Intermediate values of α lead to differentiated contributions because of the complex interactions between the fraction of wealth lost and the risk curve.”

What do you mean by differentiated contributions? Are they differentiated by amounts, rounds, players, ...?

Thank you. We meant amounts of contributions, we have corrected the sentence.

Line 89: “The reason is that intermediate potential losses imply an inherent uncertainty as the number of rounds increase —intermediate losses may accumulate over the course of the game, adding up to a large overall loss.” The reasoning for optimal α seems obscure and probably follows from the intuition the authors derive from the study they did but does not follow through the results the readers see. Perhaps the analytical treatment would clarify this.

In general, contributions increase with increasing α . However, for particular risk curves the contributions are decreasing with large α . This can happen under uncertainty at intermediate risks or α s. However, we feel that this effect is quite special and should not be the focus of our work, it has been discussed in Hilbe et al, PLOS One, 2013.

In figure 1

Some scenarios are much more risky than others as they systematically present higher probabilities of failure. Is there a way to compare scenarios that are similar? Is this how the authors classify mild and catastrophic scenarios?

Thank you. Yes, we have not assumed that risk curves are comparable in the sense that the risk across all contributions is the same – this would also immediately imply several other issues that would come up. However, we realize that there are two issues that we have not delineated enough in the previous manuscript: One is the probability that a loss occurs (given by the risk curve) and the second one is the magnitude of the loss controlled by α (small α is a mild loss and very large α a catastrophic loss). With multiple rounds, it is no longer possible to look at a single, combined parameter for this.

Line 108: To summarize, we find that for the game with multiple losses, high initial risks and full losses are not necessary to observe cooperative
Isn't this written two sentences before?

Thank you we have omitted the previous sentence.

Figure 3:

Again, either changing the x-axis to round or explicitly write round 1, round 2, ... in the different colors would make the figure more readable. For a while I thought $r=1$ meant only one round, and so on. Clearly, nothing is missing and with some effort one can interpret the figure. But all the arguments are of contributions over time and that is not what comes immediately from the image.

We agree that the previous figure 3 was too ambitious and a figure showing contributions vs. rounds is crucial for our presentation. We have now replaced Fig. 3 by such a figure which is hopefully easier to interpret. The previous figure is now part of the supplementary, Fig S2.

Line 118: the game with multiple losses elicits a similar amount of contributions as the game where the potential loss only occurs in the first round
Not for the threshold game with those parameters. The authors could clarify what they mean by similar. There are similarities and differences between all the scenarios. It would be important to clarify what those similarities are.
If the authors are comparing total contribution, it would be good to have a cumulative contribution line to really compare the scenarios in what total contribution is concerned.

We agree that this has to be clarified, we now added further explanation.

Line: 123: It seems that if risk is in a fixed round and exhibits strong threshold effects, individuals can time their contributions accordingly. However moving beyond this particular case, we find that an "act quickly" strategy is most robust. It seems that according to what?

Thank you we have fixed the sentence.

Conclusions: Line 131: how rapidly losses increase with the intensity of the events Aren't losses and intensity of an event the same thing over all the models that we see in the literature? As I see it, one assumes losses as a fundamental

quantity in the decision making, i.e., the intensity of an event is the losses it causes to a single individual. Could the authors clarify?

The idea that alpha puts on the table is that not all risky or disastrous events would affect individuals the same way. Here we aim to disentangle the probability and intensity of losses.

Line 156: We have shown that under uncertainty in terms of the timing of losses, individuals increase their efforts to ensure contributions are sufficient not only for now, but that their future is also protected. Importantly, our results provide another compelling argument for immediate climate action.

I find this argument a little confusing. The model is not descriptive of the actions of individuals, right? The model finds from an EGT point of view the strategies that are most fit to deal with the game that is posed to the players. Otherwise, one could ask “why most nations do not contribute to climate mitigation?”. The last sentence, on the other hand, does make sense in the light of the interpretation that the model is capturing the best unilateral responses if we agree the game is the one being played by the actors. I feel the manuscript would have a lot to gain if this argument is clear from the beginning and one leaves no doubt of what the model is trying to capture: is it describing the behavior of players over time when they face this dilemma or is it finding the patterns of contributions that are able to persist when players are adopting the most successful behaviors in a rationally bounded environment?

We agree that the phrasing of the end of our conclusion was unlucky. We now focus on what the model tells us: Which strategies would be evolutionary advantageous in such a scenario. The last sentences have been reworded accordingly.

Methods:

Line 162: We use evolutionary game theory to identify robust contributions between interacting players. Robust over time, you mean? Or are we talking about “evolutionary robust strategies”?

Clarified. We refer to evolutionary robustness here.

Line 163: This means players do not know the structure of the game and cannot apply advanced reasoning about their possible actions. Exactly, which seems to contradict the definition of stability in line 73.

Not necessarily, as evolutionary stability implies stability in terms of Nash

equilibria here. However, we reworded our notion of stability in the previous line 83, as we start from evolutionary stability.

170: [Is $a > b$ or vice versa? What is the strategy that emerges?]

Line 169: that depends on the collective contributions accumulated over all rounds so far, C_r

The threshold depends on the contributions? Or on the round? I thought τ_r was a set of numbers that defined the strategy of a player. Is this correct?

We have now added an example of a simple 2 round game with two players and their strategies to make sure that our explanation is understandable. We do not ask for $a > b$ or $a < b$, we just say that there is a different threshold τ of the total contributions so far in each round r and that players invest a certain amount below it and another one above it. Restricting the model to $a < b$ would mean that players would invest more if more has already been contributed, which seems a strong restriction to us.

Line 179: Errors occur with a probability ϵ for the thresholds τ and the contributions of each round independently. If they occur, errors in the threshold values add Gaussian noise with standard deviation σ to them. Errors in the contributions are made using a uniform distribution.

It would improve reproducibility if the authors clarify if errors occur in the reproduction process or during the game. Also, the uniform distribution is between what and what? 0 and W_0 ? Can players end the rounds with negative endowment? Other: There is a lost full stop in line 167.

Players cannot spend more than what they have, we do not allow negative payoffs. We have clarified this and the other comments in the text.

Notes on average Payoff

[...]

We highly appreciate the efforts of the reviewer to put our analysis on a more solid, analytical basis. A crucial part of the model is that the probabilities p depend on the past actions, which implies an intricate entanglement of the dynamics between several players in the case of heterogeneous strategies. The approach the reviewer outlines seems to be suitable for an analysis in the spirit of adaptive dynamics and could serve to analyze our model in much more depth in the future. For now, we have applied our general frame work (Hagel et al., Scientific Reports 2016) to this boarder setup that includes heterogeneity in

wealth. Please refer to the analytical results section in the supplementary material.

Reviewer #2 (Remarks to the Author):

Our understanding of the underlying mechanisms responsible for promoting and maintaining cooperation within the context of a collective-risk social dilemma (applicable to the prevention of climate change scenarios) is limited to some extent, despite many excellent papers appearing in the related literature in recent years. This paper continues this line of work by presenting an analysis of an extension to the collective-risk social dilemma, using evolutionary game theory. As such, the paper represents a small incremental study, extending earlier work from some of the authors (eg., [18] [19]), rather than a describing novel research direction.

Thank you, we agree that *“our understanding of the underlying mechanisms responsible for promoting and maintaining cooperation within the context of a collective-risk social dilemma (applicable to the prevention of climate change scenarios) is limited”* and we hope that our expansion of this game broadens our perspective and understanding. In this study we extend the typical game, which only considers a single complete loss event, to other scenarios where player can potentially retain some wealth despite multiple losses. Now, in this resubmission, we also include the effects of inequality (rich and poor) and how multiple rounds and loss events affect their contribution - this is a closer application to the prevention of climate change, but of course remains within the restrictions of our model. We have an additional supplementary file describing the results of the simulations.

The authors explicitly examine the perception of risk, and associated implications when making a decision (contribution), by focussing on specific risk curves and timing of `disturbances' (of varying severity?) in the game. They develop a game where individuals are faced with the risk of a loss in a specific round of the game, which can be a `one-off' single round occurrence or could occur in random round of the game. Exploring the effects of a potential losses in this setting is interesting, especially when individuals can retain some of the resource after an `event.' This approach is reasonable given the constraints of the adopted abstract modelling framework.

Thank you.

The paper is generally well written. However, some aspects of the

approach/methodology require clarification. For example, further details describing the definition of (distinction between) `games per generation and the number of rounds in a game would be of benefit to the reader, allowing other researchers to implement the model unambiguously. Also, further clarification for the selection of risk curves should be included (was it based solely on previous work?).

Thank you, we have clarified this now in response to the three reviews.

Detailed simulation results and analysis have been included in the paper. The metrics used for evaluation were defined clearly. The results reported in Fig 2-3 are 'steady-state' values (?). Representative temporal dynamics could also be included, illustrating the effectiveness of the learning process (particularly in terms of risk curves and the `severity of any loss.'

Yes, we have shown the steady state before. To reduce the resulting confusion, we have now replaced the previous figure 3 by the new figure 2, showing the (average) temporal dynamics.

Additional discussion of the effects of parameter values (as depicted in Fig 3) is warranted. Also, there is room to explain/justify the appropriate similarity between any risk curve for large λ . More generally, why does a `power-function lead to results that appear to be significantly different to `step-like' functions across λ values? (see Fig 3a Fig 3b)

The step function originally used by Milinski et al 2008 was discontinuous, individuals experience a zero risk or a 90% risk, for instance. Discontinuous functions are captured in our case only in the extreme limit as λ increases to infinity for the Fermi curve. Since this discontinuous case has been the main focus in the past, we aimed to explore how this special assumption may drive results.

We have also changed our notation slightly, as it was misleading to use the same λ to control the shape of very different curves. In the limit of large λ , all these curves have very different properties.

On the whole, the results and analysis are appropriate, supporting the claims made in the text. The conclusion is appropriate, and has been discussed in the wider context of coupled human-social-ecological systems.

Thank you.

Additional minor comments:

Does stability within the model (see lines 73-75) always emerge (under all circumstances)?

Since our model is stochastic and includes mutations, this question is not trivial to answer. One would expect a stationary distribution, as our evolutionary process corresponds to a recurrent Markov chain. For low mutation rates (as we consider), one expects that the stationary distribution puts most weight on states that are stable in the case of no mutations.

What impact does the shape of the risk curve actually have? Can a clear relationship in terms of λ be stated?

No. We were interested in exploring very different curves and these do not lead to a single curve in a limit. Thus, the shape of the curve has to be explored separately and we apologize for the somewhat confusing notation in the previous submission. However, from a single round game we know that strongly declining risk can stabilize distribution and that the decrease in the risk curve has to be related to the number of players.

What is the analytical prediction for the game outcome over multiple rounds? How does this depend on the risk curve/parameter values?

We feel that such a detailed analysis of our model would only make sense if the basic model is accepted by the community and is thus beyond the current scope of the work. So far, there is a lot of focus on either the standard public goods game or the game with a strong threshold effect and a single risk in the last round. Moreover, the standard analytical predictions do not take into account stochastic effects and it is known that these can have a major impact on the results.

System dynamics are guided by the embedded 'social learning' /evolutionary updated mechanism. Further analysis/description would be interesting.

The evolutionary update mechanism has been used a variety of contexts and in basic settings, most evolutionary processes lead to very similar results. The update we use has the additional benefit that one can consider non-weak selection, where the results become closer to the deterministic processes that have been considered since the early days of evolutionary game theory.

How does the unpredictable nature of a 'catastrophic disturbance' impact system dynamics over (a) longer time periods/rounds; (b) larger population sizes; and (c) the value of α for these different parameter values?

We focus on steady states after many generations and large populations, which is typically the most robust case. We assume that the reviewer is interested in longer games (i.e. more rounds) with more players instead. We have explored these issues in previous publications with a related setup, but a much simpler risk scenario (e.g. Abou Chakra et al., PLOS CB, 2012). We thus decided to focus on the simpler case of two players and up to four rounds here, as our parameter space is very high dimensional.

There is room to clarify the captions in Fig 1 and 2

Thank you, agreed. We have rewritten the first caption and moved Fig. 2 to the appendix.

* Fig 3 suggests that there is significant difference in the performance metric (av contribution) for specific risk curves, λ values, with magnifications of differences in the early round. The summary of results is consistent with the plots, however, what happens over longer time periods / rounds, especially for larger λ values? For small λ values, I am not convinced that a general conclusion can be made (based on the evidence presented). Further theoretical analysis and supporting numerical simulations would help to clarify these points.

Thank you, we hoped to ensure consistency across. We presented the first 4 round because we are confident that our results can be generalized to more rounds. However, in the interest of continuity and readability, we did not venture into these cases. We now have an additional supplementary figure S3 to show this, going up to 8 rounds.

From line 117 ... "Interestingly, the game with multiple losses elicits a similar amount of contributions as the game where the potential loss only occurs in the first round" ... Further discussion/clarification would enhance the paper.

We have reworded this statement and replaced the associated figure.

From line 149-150 "This is a striking difference between our current and

previous research on climate cooperation".... Perhaps you could highlight the fact that the model described in this paper represents a small incremental step.

Thank you, but we politely disagree here. In our opinion, the current focus on models with extreme thresholds and a single risk in the very end can be misleading and the consideration of continuous risk curves was a first step beyond that particular case. Recurrent losses could lead to very different conclusions about the mitigation of climate change, which should be taken into account carefully by researchers in this area.

Reviewer #3 (Remarks to the Author):

Nature Communications Review: "For uncertain climate change, immediate action is the best strategy"

The authors use evolutionary game theory to model a collective-risk game wherein individuals may suffer losses of varying degrees over one to four rounds of play. The authors examine the effects of the timing, probability, and extent of damages on average contributions using simulations of a model developed specifically to look at the question of whether delayed losses affect the contribution profiles of individuals in the game.

The paper touches on a few important aspects of climate change that have not been sufficiently covered in the (experimental) literature on international climate cooperation (e.g., Milinski et al., 2008; Milinski et al., 2011; Tavoni et al., 2011; Dannenberg et al., 2014). For instance, the uncertainty of the timing of losses and the possibility of recurring losses are both important features of the climate problem that require further scrutiny to enhance the generalizability of the existing literature (which is more or less confined to theoretical and experimental investigation due to the difficulty of implementing field studies on the subject). Additionally, the cumulative effects of contributions toward reducing future risk represent a realistic component of the climate problem often neglected in the literature.

While the authors appear to situate the paper within the experimental literature on international climate cooperation (see reference to Milinski et al., 2011; line 43), the study presented here uses evolutionary game theory to draw conclusions regarding individuals' responses to various features of climate damages. However, these models may produce questionable predictions, as evidenced through comparison with observed outcomes. For example, Smead et al.'s (2014) finding that heterogeneity of endowments (wealth) actually increases

cooperation goes against both intuition and all theoretical and experimental results of which I am aware (e.g., Barrett, 2004; Tavoni et al., 2011), and other conclusions drawn in the Smead paper (e.g., regarding restricting private demands) were not replicable in a lab experiment designed to closely model their ABM (Gosnell and Tavoni, 2017). Therefore, perhaps aside from its specific application here, I am rather skeptical of the method's ability to predict actual behavior amidst convoluted incentive structures outlined in the behavioral economics and political economy literatures.

Thank you for these comments and the references. We actually think they strengthen our approach and do not see contradiction between our theoretical results and what Smead et al 2014 or Gosnell and Tavoni, 2017 show in their paper. To explain: our research, herein and our previous work (especially Hilbe, et al Plos One 2014 and Abou Chakra & Traulsen 2014 JTB) as well as Smead et al 2014 and Gosnell and Tavoni, 2017 all point to a similar trend - an increasing round number increases cooperation.

1) the trend shown in Gosnell and Tavoni, 2017 Table 2 SYM and ASYM experiments (others cannot be compared -- we cannot model side negotiations/talk in our theory) show just that, and increase in cooperation as rounds increase.

2) Smead et al 2014 also shows the same trend if one looks at Figure 1. This resembles closely 3) our round effects findings in Hilbe, et al Plos One 2014, cf. figure 3a.

The current paper also shows this trend, increasing round number increases the amount individuals contribute and the timing. Fig 2-3 herein. We have improved our discussion of this issue now.

Moreover, the usefulness of the main findings themselves may be limited. The authors claim that the “act quickly strategy is most robust due to uncertainty”, which simply endorses adoption of the precautionary principle in a situation where the necessity of early action is widely accepted. The existing question is not one of whether acting quickly is the correct strategy in the presence of multiple dimensions of uncertainty; instead, it is how to successfully promote and achieve such near-term cooperation (i.e. it is a political economy / human behavior problem). What happens when we play this game with actual people, and with (many) heterogeneous players?

We cannot claim that we offer directly applicable insights about the political economy and human behavior problem. In our opinion, the most important insight of our work is that the different risk scenarios do have potentially important

consequences and should be explored further by theoretical and experimental work. We illustrate this by showing that evolutionary dynamics/social learning can lead to different outcomes.

We agree that our homogeneous scenario where all players are identical is limited, thus we have run further simulations to prove that this will work under wealth heterogeneity. Please see the revised text and supplementary material, where we (i) explore the effects of heterogeneity and lost fraction α and (ii) the timing of contributions in rich and poor players. We can show that our conclusions of early contributions of “act quickly” still holds, even when players are heterogeneous in wealth and in the amount of the wealth they will potentially lose.

It seems to me this paper could be substantially enhanced if the authors complement the model with a behavioral experiment that allows for direct comparison of the results from their simulations to observed outcomes from complex and non-deterministic human interactions; the outcomes of both methods may then be compared and contrasted with established economic theories and various relevant real-world contexts (especially international climate negotiations) to comment on generalizability. The experimental addition would not only allow the research to avoid critique regarding the general oversight of AB models' explanatory performance (e.g., see Windrum et al., 2007), it would also provide methodological insight and a means for direct comparison of the outcomes of the model with the experimental literature cited in the paper's introduction. Furthermore, for the sake of replicability and transparency, it would be a good idea for the authors to make their source code available with the publication, and perhaps also provide (supplementary) information with regards to the model's sensitivity to the chosen parameters.

Thank you we now have added a supplementary material that further clarifies our approach. We also share our program in a git hub repository (<https://github.com/abouchakra/Collective-Risk-Dilemma>). However, we are not in a position to change our theoretical work into an experimental study of behavior in such a game. None of us is an expert in experimental design and behavior and we feel that this would be beyond the scope of the current study to turn a theoretical paper into an experimental one. In fact, this would most probably provide sufficient material for a separate paper.

Minor edits:

- “Reoccur” and “reoccurrence” should be replaced with “recur” and “recurrence” throughout the manuscript

Done.

- First sentence of abstract: Climate change will not be mitigated; perhaps change to “dangerous climate change”? (And “requires” global cooperation.)

The term “dangerous climate change” is sometimes seen as a technical term for threshold driven, irreversible climate change. However, we have reworded the sentence.

- o First sentence of paper: is it true that “everyone” needs to exert effort to reduce climate change effects? Consider rephrasing this sentence.

Agreed and changed.

- Change the comma after “debatable” to a semicolon.

Done.

- Line 23: decide “whether” to contribute (remove “or not”).

Thank you, we adjusted the sentence.

- Lines 24-25: “But in light of dangerous climate change...” Why? Due to the framing of losses/damages rather than benefits accrued from the public good? Justify this claim.

Milinski et al 2008 devised a game framed around losses and damages to bring about the importance of the climate effect. Moreover, the temporal aspect of negotiations, thresholds and partial returns were important components of these studies.

- Lines 39-40: commas after “Also” and after “will be”

Thank you we adjusted the sentence accordingly

- Lines 42-44: this sentence needs some restructuring or grammatical editing.

Thank you we adjusted the sentence accordingly

- Lines 51-97: These are not results nor discussion, this is your model; perhaps

create a section for this part that precedes the results section.

We have now changed this and hope it is consistent with the formatting requirements of the journal.

- Line 114 (and subsequent text): How is the overall probability of losses held constant when losses occur either one time or multiple times? This was not clear to me from the text in the manuscript, and further explanation may be merited.

We do not keep the overall expected losses constant here, this would require to e.g. reduce alpha when more losses can occur. However, as we are mostly interested in the timing of contributions, this also does not seem to be necessary.

- Line 167: remove the full stop after 'strategy' (and add one to line 184 after 'dynamics').

Thank you we adjusted the sentence accordingly.

We would like to thank all three referees for their detailed, insightful and constructive criticism on our previous manuscript.

Reviewers' comments:

Reviewer #1 (Remarks to the Author):

I appreciate the attention the authors gave to addressing the questions raised. In particular, that they moved the focus away from the different curves and that they now make a clearer distinction between behavior and strategy. Also, the new figure 2 really helps reading the paper.

The authors introduced a new layer of complexity on the model (which they managed by reducing the number of rounds). I do like this idea and it seems a good complement to the paper even though, as it is, it looks tangential. The Abstract does not mention any inequality testing; apart from reference 17, there is nothing in the Introduction either. The last three words of the Conclusions is the only reference to heterogeneity and it not clear if the authors mean heterogenous endowments. The Methods also don't mention rich and poor (with W_{rich} and W_{poor} being undefined in the text, as far as I could tell). Clearly, I think the authors should mention they tested this too and articulate the text such that it is clear what and why they did and not just something that was added afterwards to have an analytical treatment in the paper.

Regarding the description of the model, I was wondering if there is any figure or analysis with a group larger than 2. It is not clear as in line 80 the authors say "typically 2" but then in Figures 1,2, S1,S2, and S3 the game is between 2, and in Figure 3, S4, and S5 I believe it is also 2 players, one rich and one poor. If it is always two it would be better to make that statement without ambiguity. Additionally, I think it is very important to describe how the method changes in the heterogeneous case. What is the fraction of player of each type? I assume you restrict interactions between individuals of the same class. Is that so?

The paper is lacking a brief discussion on how it relates to the real world and climate change mitigation in particular. Currently, we are not observing the successful strategies found by the model, which indeed reduce effective risk to zero. How does this help our comprehension of the problem? As I see it, the answer lies somewhere between discounting, perception of risk but, maybe what this work shows best, perception of individual impact on risk, as the authors imply when they say "as long as individuals can make a difference" l182. Additionally, I suggest complementing with a discussion on effects that were not considered that could potentially change this result for better or worse.

Finally, equation [1] is not mentioned in the manuscript anywhere so it probably can be put inline with the text. Also, I don't understand why is the second equality helpful. I still believe that the formulation I gave in last revision is more easily interpreted and generalized for any number of rounds: $w_i = (1-p_2 \alpha)((1-p_1 \alpha)(W_0 - c_{(i,1)} - c_{(i,2)}))$, i.e., players keep a fraction $1-p_1 \alpha$ of whatever they expect to have in the previous round after contributing.

I believe that after the authors harmonize the newly introduced analysis of heterogenous players with the remaining text and improve their conclusions with a critique of the model the paper can be considered for publication.

Minor:

Line 42: "predictions when" -> predicting when

Line 43: "it will hit or how drastic it", it is not clear what it references

Line 44: "were" -> "where"

Line 45: "asses" -> assess

Line 60: M is not defined and it seems to be always 2.

Line 69: "expected expected"->expected

Line 100: "The effects of timing have only recently become apparent, experimental evidence and theoretical framework all point to the trend that..."->"The effects of timing have only recently become apparent, with experimental evidence and theoretical framework all pointing to the trend that..."

Line 115: "a risk curves" -> "a risk curves"

Line 167: The paragraph that starts has a weird articulation. In particular, the sentences "The collective-risk game...a high loss probability promotes contributions" seem a little lost. Their

position makes the text alternate between the authors' model, literature, and again model and results. Could easily be improved for readability.

Line 200: "which are" -> "which is". The strategy is singular in the previous clause.

Line 213: "in the two player game, these groups are pairs of players". Either I don't understand what you mean or this is a very redundant sentence. "two player" should be hyphenated.

SI 120: "Othe"->the

SI 127+28 and 134+35 are exactly the same and the part "their wealth they will lose" sounds weird.

SI 150: remove "?", as it is not a question but a statement of the question.

SI 69: reference of the table is gone.

SI the notation for the endowment is slightly inconsistent in the Manuscript and SI. Sometimes the authors use W_0 , sometimes W , or W_{rich} , W_{R0} , W_R , and equivalently for the poor.

Figure S6: why would the rich have higher risk than that of the poor?

Table 1: It would be interesting to have a plot of c_R^*/c_P^* as a function of w_R/w_P (for $w_R/w_P > 1$) for a few λ s, comparing to the 1:1 line, showing that, in a rich vs. poor interaction, it would be in the self-interest of the rich to overcarry their weight as inequality increases.

Reviewer #2 (Remarks to the Author):

The authors responses, edits to the paper (clarifications and/or re-drafting where necessary) have clarified my initial concerns.

I have worked through the paper very carefully. I am confident that the paper presents an important contribution to the domain.

I am now recommending that the paper be accepted for publication.

Reviewer #3 (Remarks to the Author):

Reviewer #3: R&R Comments for "Facing uncertain climate change, immediate action is the best strategy"

The authors have improved the readability of the paper and situated it better within its literature, though in my view there could still be more insightful discussion surrounding the crucial caveats with respect to political economy and behavioral constraints, as well as better clarification regarding the selection of parameters to feed into their model. Additionally, varying overall expected losses removes the ability to make direct comparisons between the 'potential losses in all rounds' game and the 'losses occur in one random round' game (e.g., lines 129-130). With regards to overall writing mechanics, considerable editing and proofreading is required before this paper should be published, and the authors should be careful not to neglect commentary with regards to tables and figures (e.g., there is no commentary at all in section 2.1 of the Supplementary Material). As mentioned above, justification for selection of parameters (e.g. group size, number of rounds considered, proportional differences in wealth between R and P players, etc.) would help the reader to feel as though there is purpose behind these otherwise seemingly arbitrary numbers (though ideally these choices would have been pre-registered to assure us that these choices have not been subject to various researcher degrees of freedom).

Minor comments:

- 'Uncertain' appears twice in the title
- Line 13: remove 'that'
- Line 15: what is 'its' referring to?
- Line 20: replace 'needs the' with 'requires'

- Line 21: replace 'on' with 'at', remove comma after 'challenge'
- Line 23: 'describing that the' doesn't make sense – revise wording
- Line 40: 'draughts' should be 'droughts'
- Lines 44-45: 'were' should be 'where', 'asses' should be 'assess'
- Line 66: 'the' should be 'then'
- Line 68: remove 'always'; unless I'm missing something, $0 \leq c_{i,r} < W_{i,r-1}$ is a constraint here, not an assumption
- Model section: Need to be clear that W is equal across players in your initial model, i.e. players are homogeneous (otherwise socially optimal behavior stated in lines 75-76 may not hold)
- Line 69: remove 'expected' (appears twice)
- Line 72: Remove parentheses (you've already stated this constraint above)
- Line 80: I am still uncertain why you only feed into the model groups of size 2; it would be good to get some justification here.
- Line 115: should say 'for risk curves with a threshold'
- Line 130: 'with' should be 'where'
- Line 137: 'occur' should be 'occurs'
- Line 140: this is a run-on sentence – surround the middle clause with dashes, perhaps.
- Line 144: 'disaster' should be singular
- Figure 3: Again, some explanation for the selection of parameters might be useful – seems somewhat arbitrary at the moment.
- Lines 171-177: These sentences appear to come out of nowhere; I suggest deleting these, and moving the following sentence (beginning with 'Our') to before 'Specifically' on line 168. This paragraph does not flow at present.
- Line 191: again, 'on' should be 'at'.
- Line 224: needs a reference.
- "In addition to a potential loss event in every round or only in the last round, we now consider scenarios with a potential loss only in the first round or in a single random unknown round within the game." What is this testing intuitively? For which climate scenarios is this relevant?
- Line 152: "Random unpredictable risk events" – define? Or should this say 'loss events' instead of 'risk events', or 'unpredictable-risk events'? (It seems also that 'lost fraction' alpha should instead be called a 'loss fraction')
- Supplementary: Line 45 should say 'negotiations'
- Figure S4: "It is assumed that poor players lose everything, i.e. $\alpha P = 0.5$." I think this should say poor players lose half their wealth?
- Supplementary Section 2.1 lacks any commentary?

Reviewer #1 (Remarks to the Author):

I appreciate the attention the authors gave to addressing the questions raised. In particular, that they moved the focus away from the different curves and that they now make a clearer distinction between behavior and strategy. Also, the new figure 2 really helps reading the paper.

Thank you.

The authors introduced a new layer of complexity on the model (which they managed by reducing the number of rounds). I do like this idea and it seems a good complement to the paper even though, as it is, it looks tangential. The Abstract does not mention any inequality testing; apart from reference 17, there is nothing in the Introduction either. The last three words of the Conclusions is the only reference to heterogeneity and it not clear if the authors mean heterogenous endowments. The Methods also don't mention rich and poor (with W_{rich} and W_{poor} being undefined in the text, as far as I could tell). Clearly, I think the authors should mention they tested this too and articulate the text such that it is clear what and why they did and not just something that was added afterwards to have an analytical treatment in the paper.

We agree that the addition of the rich and poor scenario was not linked well with the previous work. We have now

- (i) added a sentence in the abstract on this
- (ii) mentioned explicitly how one can address heterogeneity
- (iii) improved the link in the conclusions

Regarding the description of the model, I was wondering if there is any figure or analysis with a group larger than 2. It is not clear as in line 80 the authors say "typically 2" but then in Figures 1,2, S1,S2, and S3 the game is between 2, and in Figure 3, S4, and S5 I believe it is also 2 players, one rich and one poor. If it is always two it would be better to make that statement without ambiguity.

We agree. We had indeed only used two players in the simulations we have shown, but we aimed to avoid giving the impression that this is a fundamental restriction of our work. We now show the new figure S2 in the SI which indicates that the model is not restricted in terms of the number of interacting players, but at the same time we reworded the statement.

Additionally, I think it is very important to describe how the method changes in the heterogeneous case. What is the fraction of player of each type? I assume you restrict interactions between individuals of the same class. Is that so?

No, we have so far focused on the interaction of one rich and one poor player, i.e. the different classes play with each other. In terms of strategy spreading, however, the strategies of the rich only spread to other rich players (and equivalently for the poor).

The paper is lacking a brief discussion on how it relates to the real world and climate change mitigation in particular. Currently, we are not observing the successful strategies found by the model, which indeed reduce effective risk to zero. How does this help our comprehension of the problem? As I see it, the answer lies somewhere between discounting, perception of risk but, maybe what this work shows best, perception of individual impact on risk, as the authors imply when they say “as long as individuals can make a difference” 1182. Additionally, I suggest complementing with a discussion on effects that were not considered that could potentially change this result for better or worse.

We have now added additional discussion and literature regarding heterogeneity and the perception of risk. Please note that we are hesitant to make specific connections to behavior in the real world, since many factors influence a single decision. Our theoretical study only allows us to make a statement that the risk scenario will matter for optimal behavior, but it would be exaggerated to link it to specific behaviors.

Finally, equation [1] is not mentioned in the manuscript anywhere so it probably can be put inline with the text. Also, I don't understand why is the second equality helpful. I still believe that the formulation I gave in last revision is more easily interpreted and generalized for any number of rounds: $\pi_i = (1-p_2 \alpha)((1-p_1 \alpha)(W_0 - c_{(i,1)} - c_{(i,2)}))$, i.e., players keep a fraction $1 - p_1 \alpha$ of whatever they expect to have in the previous round after contributing.

We agree that this formulation could be more instructive if one aims at delineating the rounds and not the four possible scenarios. Thus, we now adopted the formulation proposed before already, but we prefer to have the equation more visible than an inline equation. Thank you.

I believe that after the authors harmonize the newly introduced analysis of heterogeneous players with the remaining text and improve their conclusions with a critique of the model the paper can be considered for publication.

Minor:

Line 42: “predictions when” -> predicting when

Line 43: “it will hit or how drastic it”, it is not clear what it references

Line 44: “were” -> “where”

Line 45: “asses” -> assess

Line 60: M is not defined and it seems to be always 2.

Line 69: “expected expected”->expected

Line 100: “The effects of timing have only recently become apparent, experimental evidence and theoretical framework all point to the trend that...”-
>“The effects of timing have only recently become apparent, with experimental evidence and theoretical framework all pointing to the trend that...”

Line 115: “a risk curves” -> “a risk curves”

Line 167: The paragraph that starts has a weird articulation. In particular, the

sentences “The collective-risk game... a high loss probability promotes contributions” seem a little lost. Their position makes the text alternate between the authors’ model, literature, and again model and results. Could easily be improved for readability.

Line 200: “which are” -> “which is”. The strategy is singular in the previous clause.

Line 213: “in the two player game, these groups are pairs of players”. Either I don’t understand what you mean or this is a very redundant sentence. “two player” should be hyphenated.

Thank you for the careful reading, we have implemented all the minor changes in the text.

SI I20: “othe”->the

SI I27+28 and I34+35 are exactly the same and the part “their wealth they will lose” sounds weird.

SI I50: remove “?”, as it is not a question but a statement of the question.

SI 69: reference of the table is gone.

SI the notation for the endowment is slightly inconsistent in the Manuscript and SI. Sometimes the authors use W_0 , sometimes W , or W_{rich} , W_{R0} , W_R , and equivalently for the poor.

Thank you, we have implemented all the minor changes in the supplementary text.

Figure S6: why would the rich have higher risk than that of the poor?

It has been argued that rich nations do not fear climate impact while other nations (typically poor) have had several drastic climatic events and thus consider risks in their decisions. We think it is interesting to explore this scenario, but for completeness we also included the case where the risk for the rich is higher, not lower. We have now included this in the discussion.

Table 1: It would be interesting to have a plot of c_R^*/c_P^* as a function of w_R/w_P (for $w_R/w_P > 1$) for a few lambdas, comparing to the 1:1 line, showing that, in a rich vs. poor interaction, it would be in the self-interest of the rich to overcarry their weight as inequality increases.

We have tried this and many other ways to plot the contributions, but we realized that this is complex and very hard for non-experts to interpret. We thus decided to add figure S10 to the SI with a slightly different format, where four inequality scenarios are shown that make the effect of increasing equality and changing risk very transparent.

Reviewer #2 (Remarks to the Author):

The authors responses, edits to the paper (clarifications and/or re-drafting where necessary) have clarified my initial concerns.

I have worked through the paper very carefully. I am confident that the paper presents an important contribution to the domain.

I am now recommending that the paper be accepted for publication.

Thank you, we are glad that we managed to address the earlier concerns.

Reviewer #3 (Remarks to the Author):

Reviewer #3: R&R Comments for “Facing uncertain climate change, immediate action is the best strategy”

The authors have improved the readability of the paper and situated it better within its literature, though in my view there could still be more insightful discussion surrounding the crucial caveats with respect to political economy and behavioral constraints, as well as better clarification regarding the selection of parameters to feed into their model. Additionally, varying overall expected losses removes the ability to make direct comparisons between the ‘potential losses in all rounds’ game and the ‘losses occur in one random round’ game (e.g., lines 129-130).

We agree that this comparison is not straightforward, we now make it clear that we refer to the temporal pattern of contributions only.

With regards to overall writing mechanics, considerable editing and proofreading is required before this paper should be published, and the authors should be careful not to neglect commentary with regards to tables and figures (e.g., there is no commentary at all in section 2.1 of the Supplementary Material). As mentioned above, justification for selection of parameters (e.g. group size, number of rounds considered, proportional differences in wealth between R and P players, etc.) would help the reader to feel as though there is purpose behind these otherwise seemingly arbitrary numbers (though ideally these choices would have been pre-registered to assure us that these choices have not been subject to various researcher degrees of freedom).

Thank you, we have carefully revised the manuscript to improve the connections between the different sections of the manuscript. In terms of the parameters: We have tried to use parameters (such as round number, group size and wealth) that have been discussed or used in previous models or experiments in order to improve the direct comparison. However, since this is the first time our multi-loss event game has been presented or explored we tried to explore a broad range of parameter space as shown in the text and the supplementary material. We agree that there is a large

amount of freedom in these choices, but we feel that one needs to start somewhere to encourage a careful look at such alternative risk scenarios. Moreover, in our previous work we have taken exactly this approach for an established risk scenario (see Abou Chakra & Traulsen, PLOS CB 2012) – here the exploration of a wide parameter space seems to be misplaced, first we need to convince the community to look into alternative risk scenarios.

Minor comments:

- ‘Uncertain’ appears twice in the title
- Line 13: remove ‘that’
- Line 15: what is ‘its’ referring to?
- Line 20: replace ‘needs the’ with ‘requires’
- Line 21: replace ‘on’ with ‘at’, remove comma after ‘challenge’
- Line 23: ‘describing that the’ doesn’t make sense – revise wording
- Line 40: ‘draughts’ should be ‘droughts’
- Lines 44-45: ‘were’ should be ‘where’, ‘asses’ should be ‘assess’
- Line 66: ‘the’ should be ‘then’
- Line 68: remove ‘always’; unless I’m missing something, $0 \leq c_{i,r} < W_{i,r-1}$ is a constraint here, not an assumption
- Model section: Need to be clear that W is equal across players in your initial model, i.e. players are homogeneous (otherwise socially optimal behavior stated in lines 75-76 may not hold)
- Line 69: remove ‘expected’ (appears twice)
- Line 72: Remove parentheses (you’ve already stated this constraint above)
- Line 80: I am still uncertain why you only feed into the model groups of size 2; it would be good to get some justification here.
- Line 115: should say ‘for risk curves with a threshold’
- Line 130: ‘with’ should be ‘where’
- Line 137: ‘occur’ should be ‘occurs’
- Line 140: this is a run-on sentence – surround the middle clause with dashes, perhaps.
- Line 144: ‘disaster’ should be singular
- Figure 3: Again, some explanation for the selection of parameters might be useful – seems somewhat arbitrary at the moment.
- Lines 171-177: These sentences appear to come out of nowhere; I suggest deleting these, and moving the following sentence (beginning with ‘Our’) to before ‘Specifically’ on line 168. This paragraph does not flow at present.
- Line 191: again, ‘on’ should be ‘at’.
- Line 224: needs a reference

Thank you, we have implemented all the minor changes in the text.

- “In addition to a potential loss event in every round or only in the last round, we now consider scenarios with a potential loss only in the first round or in a single random unknown round within the game.” What is this testing intuitively? For which climate scenarios is this relevant?

We now explain our reason “The random round captures, most closely, our current state since the true timing of when climate change will

lead to catastrophic losses is unknown. It seems natural to assume that individuals will behave as if climate events hit at random times.”

- Line 152: “Random unpredictable risk events” – define? Or should this say ‘loss events’ instead of ‘risk events’, or ‘unpredictable-risk events’? (It seems also that ‘lost fraction’ alpha should instead be called a ‘loss fraction’)

Thank you we used unpredictable-risk events and replaced lost fraction with loss fraction throughout the text

- Supplementary: Line 45 should say ‘negotiations’
- Figure S4: “It is assumed that poor players lose everything, i.e. $\alpha P = 0.5$.” I think this should say poor players lose half their wealth?
- Supplementary Section 2.1 lacks any commentary?

Thank you, we have implemented these minor changes in the supplementary text.

We thank all three reviewers and the editor for their constructive criticism and detailed remarks, they helped us to address a number of issues that appeared in the earlier versions and thus improved the manuscript a lot.

REVIEWERS' COMMENTS:

Reviewer #1 (Remarks to the Author):

The authors did a good job in answering the questions raised and in implementing the modifications suggested.

I think it would be useful to add a line explaining the intuition of restricting the interactions to those between different classes. As I understand, the authors are mimicking a negotiation between the group of rich and the group of poor, each represented by one player. This is a natural assumption given the natural polarization that occurs in climate change negotiations often a consequence of an historical-contribution argument.

I believe that, once the authors reread the paper carefully, they will fix all the missing reference to figures (l109, l111,l120,...), the parenthesis with full stops in the middle (l102-104), the sentences with too many adverbs, the subject and its verb separated by a single comma (final line of page 2 in SI), and they will make all the necessary final-step edits to the paper.

Reiterating the overall interest and relevance of the paper, I recommend the paper to be accepted for publication.

Reviewer #3 (Remarks to the Author):

I believe the authors have now addressed the points I raised to the extent possible and have no further comments.